# Provably Good Batch Reinforcement Learning Without Great Exploration

**Yao Liu**
Stanford University
yaoliu@stanford.edu

**Adith Swaminathan**
Microsoft Research
adswamin@microsoft.com

**Alekh Agarwal**
Microsoft Research
alekha@microsoft.com

**Emma Brunskill**
Stanford University
ebrun@cs.stanford.edu

## Abstract

Batch reinforcement learning (RL) is important to apply RL algorithms to many high stakes tasks. Doing batch RL in a way that yields a reliable new policy in large domains is challenging: a new decision policy may visit states and actions outside the support of the batch data, and function approximation and optimization with limited samples can further increase the potential of learning policies with overly optimistic estimates of their future performance. Some recent approaches to address these concerns have shown promise, but can still be overly optimistic in their expected outcomes. Theoretical work that provides strong guarantees on the performance of the output policy relies on a strong concentrability assumption, which makes it unsuitable for cases where the ratio between state-action distributions of behavior policy and some candidate policies is large. This is because, in the traditional analysis, the error bound scales up with this ratio. We show that using *pessimistic value estimates* in the low-data regions in Bellman optimality and evaluation back-up can yield more adaptive and stronger guarantees when the concentrability assumption does not hold. In certain settings, they can find the approximately best policy within the state-action space explored by the batch data, without requiring a priori assumptions of concentrability. We highlight the necessity of our pessimistic update and the limitations of previous algorithms and analyses by illustrative MDP examples and demonstrate an empirical comparison of our algorithm and other state-of-the-art batch RL baselines in standard benchmarks.

## 1 Introduction

A key question in Reinforcement Learning is about learning good policies from off policy batch data in large or infinite state spaces. This problem is not only relevant to the batch setting; many online RL algorithms use a growing batch of data such as a replay buffer [24, 28]. Thus understanding and advancing batch RL can help unlock the potential of large datasets and may improve online RL algorithms. In this paper, we focus on the algorithm families based on Approximate Policy Iteration (API) and Approximate Value Iteration (AVI), which form the prototype of many model-free online and offline RL algorithms. In large state spaces, function approximation is also critical to handle state generalization. However, the deadly triad [35] of off-policy learning, function approximation and bootstrapping poses a challenge for model-free batch RL. One particular issue is that the max in the Bellman operator may pick actions in $(s, a)$ pairs with limited but rewarding samples, which can lead to overly optimistic value function estimates and under-performing policies [26].

This issue has been studied from an algorithmic and empirical perspective in many ways. Different heuristic approaches [12, 21, 17] have been proposed and shown to be effective to relieve this weakness empirically. However the theoretical analysis of these methods is limited to tabular problem settings, and the practical algorithms also differ substantially from their theoretical prototypes.

Other literature focuses primarily on approaches that have strong theoretical guarantees. Some work considers safe batch policy improvement: only deploying new policies if with high confidence they improve over prior policies. However, such work either assumes that the policy class can be enumerated [38], which is infeasible in a number of important cases; or uses regularization with a behavior policy [37] as a heuristic, which can disallow significantly different but better policies. On the other hand, there are a number of formal analyses of API and AVI algorithms in batch settings with large or infinite state and policy spaces [29, 30, 3, 31, 8, 5, 40]. These results make strong assumptions about the distribution of the batch data, known as the *concentrability* condition. Concentrability ensures that the ratio between the induced state-action distribution of *any non-stationary* policy and the state-action distribution in the batch data is upper bounded by a constant, called the *concentrability* coefficient. This is a strong assumption and hard to verify in practice since the space of policies and their induced state-action distributions is huge. For example, in healthcare datasets about past physician choices and patient outcomes, decisions with a poor prognosis may be very rare or absent for a number of patient conditions. This results in a large concentration coefficient, and hence existing performance bounds [29, 30, 5, 3] end up being prohibitively large. This issue occurs even if good policies (such as another physician's decision-making policy) are well supported by the dataset.

In the on-policy setting, various relaxations of the concentrability assumption have been studied. For example, some methods [15, 16, 1] obtain guarantees scaling in the largest density ratio between the optimal policy and an initial state distribution, which is a significantly milder assumption than concentrability [34, 1]. Unfortunately, leveraging a similar assumption is not straightforward in the fully offline batch RL setting, where an erroneous estimate of a policy's quality in a part of the state space not supported by data would never be identified through subsequent online data collection.

**Our contributions.** Given these considerations, an appealing goal is to ensure that we can output the best possible policy which is well supported by our dataset in terms of its state-action distribution. We achieve this goal by leveraging the idea of *pessimism in face of uncertainty*. Rather than assuming concentrability on the entire policy space, we algorithmically choose to focus on policies that satisfy a bounded density ratio assumption akin to the on-policy policy gradient methods, and successfully compete with such policies. Our methods are guaranteed to converge to the approximately best decision policy in this set, and our error bound scales with a parameter that defines this policy set and controls the amount of pessimism. If the behavior data provides sufficient coverage of states and actions visited under an optimal policy, then our algorithms output a near-optimal policy. In the physician example and many real-world scenarios, good policies are often well-supported by the behavior distribution even when the concentrability assumption fails.

Many recent state-of-the-art batch RL algorithms [12, 21, 17] do not provide such guarantees and can struggle, as we show shortly with an illustrative example. Our methods use pessimistic value estimates for state-action pairs with insufficient data in the Bellman operators. The key insight is that prior works also add pessimism but only based on the conditional action distribution; we instead select among actions which have good coverage according to the marginalized support of *states and actions* under the behavior dataset, and enforce pessimistic value estimates otherwise. Our policy iteration algorithm provides the desired "doing the best with what we've got" guarantee, and we provide a slightly weaker result for a value iteration method.

We then validate a practical implementation of our algorithm in a discrete task and some continuous control benchmarks. It achieves better and more robust performance with how exploratory the data distribution is, compared with baseline algorithms. This work makes a concrete step forward on providing guarantees on the quality of batch RL with function approximation.

## 2  Problem Setting

Let $M = <\mathcal{S}, \mathcal{A}, P, R, \gamma, \rho>$ be a Markov Decision Process (MDP), where $\mathcal{S}, \mathcal{A}, P, R, \gamma, \rho$ are the state space, action space, dynamics model, reward model, discount factor and distribution over initial states, respectively. A policy $\pi : \mathcal{S} \to \Delta(\mathcal{A})$ is a conditional distribution over actions given state. To simplify the exposition, our derivations will assume that $\mathcal{A}$ is discrete – the algorithm can

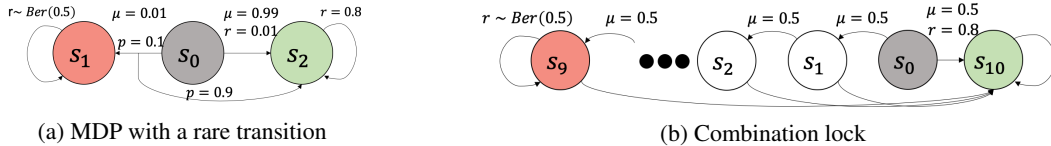

|  (a) MDP with a rare transition  |  (b) Combination lock  |

Figure 1: Challenging MDPs for prior approaches. In both MDPs, the episode starts from $s_0$ and ends after a fixed horizon, 2 or 10 respectively. The optimal policy is to reach the green state. Transition probabilities are labeled as $p$ on edges or are deterministic. $\mu$ is the conditional probability of action given state when generating the data. The reward distribution is labeled on the edges unless it is 0.

be generalized to continuous $\mathcal{A}$ straightforwardly and demonstrated in Section 6.2. A policy $\pi$ and MDP $M$ together induce a joint distribution over trajectories: $s_0, a_0, r_0, s_1, a_1, \ldots$, where $s_0 \sim \rho(\cdot)$, $a_h \sim \pi(s_h), r_h \sim R(s_h, a_h), s_{h+1} \sim P(s_h, a_h)$. The expected discounted sum of rewards of a policy $\pi$ given an initial state $s$ is $V^\pi(s) = \mathbb{E}\left[\sum_{h=0}^\infty \gamma^h r_h\right]$, and $Q^\pi(s, a)$ further conditions on the first action being $a$. We assume for all possible $\pi$, $Q^\pi(s, a) \in [0, V_{\max}]$. We define $v^\pi$ to be the expectation of $V^\pi(s)$ under initial state distribution. We are given a dataset $D$ with $n$ samples *drawn i.i.d. from a behavior distribution* $\mu$ over $\mathcal{S} \times \mathcal{A}$, and overload notation to denote the marginal distribution over states by $\mu(s) = \sum_{a \in \mathcal{A}} \mu(s, a)$. Approximate value and policy iteration (AVI/API) style algorithms fit a $Q$-function over state, action space: $f : \mathcal{S} \times \mathcal{A} \to [0, V_{\max}]$. Define the *Bellman optimality/evaluation operators* $\mathcal{T}$ and $\mathcal{T}^\pi$ as:

$$(\mathcal{T}f)(s, a) := r(s, a) + \gamma \mathbb{E}_{s'}\left[\max_{a'} f(s', a')\right] \text{ and } (\mathcal{T}^\pi f)(s, a) := r(s, a) + \gamma \mathbb{E}_{s'} \mathbb{E}_{a' \sim \pi} f(s', a').$$

Define $\widehat{\mathcal{T}}$ and $\widehat{\mathcal{T}}^\pi$ by replacing expectations with sample averages. Then AVI iterates $f_{k+1} \leftarrow \widehat{\mathcal{T}} f_k$. API performs policy evaluation by iterating $f_{i+1} \leftarrow \widehat{\mathcal{T}}^{\pi_k} f_i$ until convergence to get $f_{k+1}$ followed by a greedy update to a deterministic policy: $\pi_{k+1}(s) = \arg\max_a f_{k+1}(s, a)$.

## 3   Challenges for Existing Algorithms

Value-function based batch RL typically uses AVI- or API-style algorithms described above. A standard assumption used in theoretical analysis for API and AVI is the *concentrability* assumption [29, 30, 5] that posits: for *any* distribution $\nu$ that is reachable for some non-stationary policy, $\|\nu(s, a)/\mu(s, a)\|_\infty \le C$. Note that even if $C$ might be small for many high value and optimal policies in a domain, there may exist some policies which force $C$ to be exponentially large (in the number of actions and/or effective horizon). Consider a two-arm bandit where the good arm has a large probability under behavior policy. Intuitively it should be easy to find a good policy supported by the behavior data but there exist some policies which are neither good nor supported, and the theoretical results requires setting an upper bound $C$ for admitting policies choosing the bad arm. [1]

Nevertheless, the standard concentrability assumption has been employed in many prior works on API [22, 29] and AVI [30, 31, 10, 5]. We also observe that without algorithmic changes, the poorly supported, low-value policies have high variance estimates in the example scenario above, and can indeed be returned by AVI/API erroneously, suggesting that the assumption is not superfluous, but rather correctly captures the behavior of these methods. As further evidence, these algorithms often diverge [33, 14], potentially due to the uncontrolled extrapolations caused by both the failure of this assumption and function approximation. Such empirical observations have helped motivate several recent AVI (e.g. [12, 17]) and API (e.g. [21]) works that improve stability and performance.

One attempt in this direction follows an intuition of constraining the backups of $Q$-values from certain state action pairs, and yields significant empirical improvement in several batch RL tasks. BCQL algorithm [12] only bootstraps value estimates from actions with conditional probabilities above a threshold under the behavior policy[2]. BEAR algorithm [17] uses distribution-constrained backups as its prototype for theoretical analysis which, in the tabular setting, is essentially same as BCQL using non-zero threshold. However, we find that these algorithms have failure modes even in simple MDPs shown in Figure 1, due to the fact that the constraint in their algorithm is on $\mu(a|s)$ which cannot

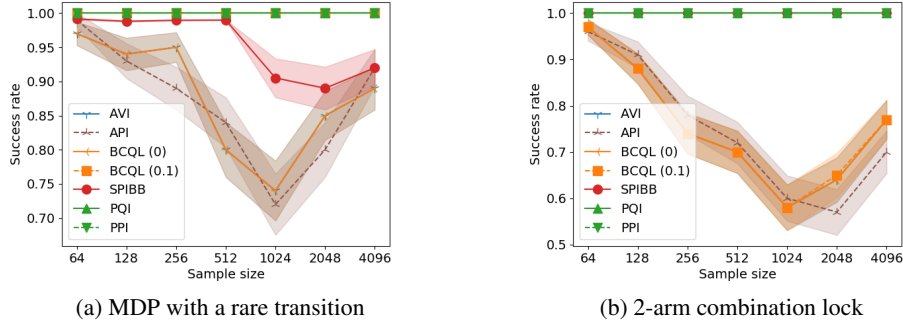

(a) MDP with a rare transition                      (b) 2-arm combination lock

Figure 2: Frequency of converging to $\pi^\star$ (Success Rate) for different variants of approximated value iteration: AVI, API, BCQL [12] (with different thresholds), SPIBB [21], our algorithms (PPI, PQI). Tabular algorithm of BEAR [17] is same as BCQL with non-zero threshold. Frequencies are computed from 100 runs, and the change of frequencies over sample size is shown across $X$-axis. Error bars are standard deviation. We use a MLE estimate as $\widehat{\mu}$ and $b = 10/\text{sample size}$.

model the whole uncertainty in the state-action space and backups. Specifically, BCQL and BEAR perform bootstrapping based on *just the action probability*, even if the state in question itself is less explored.[3] Figure 1b shows an example where a sequence of large action probabilities can result in an exponentially small marginalized state visitation. This causes BCQL and BEAR to bootstrap values from infrequent states in the data, leading to bad performance as shown in Figure 2b.

SPIBB [21] follows the behavior policy in less explored state-action pairs while attempting improvement everywhere else, assuming they know the behavior policy. Thus it is robust to this failure mode when the behavior policy is known. In this paper, we do not make this assumption as it is unknown in many settings (e.g., physicians' decision making policy in a medical dataset). Consequently, following the estimated behavior policy from rare transitions is dangerous as our estimates can also be unreliable there. This can yield poor performance as shown in Figure 2a. In that plot, rare events start to appear as sample size increases and baseline algorithms begin to bootstrap unsafe values. Hence we see poor success rate as we get more behavior data. As samples size get very large all algorithms eventually see enough data to correct this effect. Next we describe the design and underlying theory for our new algorithms that successfully perform in these challenging MDPs (see Figure 2).

## 4  Pessimistic Policy Iteration and $Q$ Iteration Algorithms

Our aim is to create algorithms that are guaranteed to find an approximately optimal policy over all policies that only visit states and actions with sufficient visitation under $\mu$. In order to present the algorithms, we introduce some useful notation and present the classical fitted $Q$ iteration (FQI) [36] and fitted policy iteration (FPI) [3] algorithms which are the function approximation counterparts of AVI and API respectively. For the FPI algorithm, let $\Pi \subset (\mathcal{S} \to \Delta(\mathcal{A}))$ be a policy class. Let $\mathcal{F} \subseteq (\mathcal{S} \times \mathcal{A} \to [0, V_{\max}])$ be a $Q$ function class in both FQI and FPI. We assume $\Pi$ and $\mathcal{F}$ are both finite but can be very large (error bounds will depend on $\log(|\mathcal{F}||\Pi|)$). For any function $g$ we define shorthand $\|g\|_{p,\mu}$ to denote $(\mathbb{E}_{(s,a)\sim\mu} g(s,a)^p)^{1/p}$. FQI updates $f_{k+1} = \arg\min_{f\in\mathcal{F}} \|f - \widehat{\mathcal{T}} f_k\|^2_{2,\mu}$ and returns the greedy policy with respect to the final $f_k$ upon termination. FPI instead iterates $f_{i+1} = \arg\min_{f\in\mathcal{F}} \|f - \widehat{\mathcal{T}}^{\pi_k} f_i\|^2_{2,\mu}$ until convergence to get $f_{k+1}$ followed by the greedy policy update to get $\pi_{k+1}$. When a restricted policy set $\Pi$ is specified so that the greedy policy improvement is not possible, weighted classification is used to update the policy in some prior works [9]. Since both AVI and API suffer from bootstrapping errors in less visited regions of the MDP even in the tabular setting, FQI and FPI approaches have the same drawback.

We now show how to design more robust algorithms by constraining the Bellman update to be only over state action pairs that are sufficiently covered by $\mu$. Implementing such a constraint requires access to the density function $\mu$, over an extremely large or infinite space. Given we have samples of this distribution, several density estimation techniques [25, 27] can be used to estimate $\mu$ in practice, and here we assume we have a density function $\widehat{\mu}$ which is an approximate estimate of $\mu$. In the analysis section we will specify how our error bounds scale with the accuracy of $\widehat{\mu}$. Given $\widehat{\mu}$ and a

| **Algorithm 1** Pessimistic Policy Iteration (PPI) | **Algorithm 2** Pessimistic $Q$ Iteration (PQI) |
|---|---|
| 1: **Input:** $D, \mathcal{F}, \Pi, \widehat{\mu}, b$ | 1: **Input:** $D, \mathcal{F}, \widehat{\mu}, b$ |
| 2: **Output:** $\widehat{\pi}_T$ | 2: **Output:** $\widehat{\pi}_T$ |
| 3: **for** $t = 0$ **to** $T - 1$ **do** | 3: **for** $t = 0$ **to** $T - 1$ **do** |
| 4:     **for** $k = 0$ **to** $K$ **do** | 4:     $f_{t+1} \leftarrow \arg\min_{f \in \mathcal{F}} \mathcal{L}_D(f; f_t)$ |
| 5:         $f_{t,k+1} \leftarrow \arg\min_{f \in \mathcal{F}} \mathcal{L}_D(f, f_{t,k}; \widehat{\pi}_t)$ | 5:     $\widehat{\pi}_{t+1}(s) \leftarrow \arg\max_{a \in \mathcal{A}} \zeta \circ f_{t+1}(s, a)$ |
| 6:     **end for** | 6: **end for** |
| 7:     $\widehat{\pi}_{t+1} \leftarrow \arg\max_{\pi \in \Pi} \mathbb{E}_D[\mathbb{E}_\pi[\zeta \circ f_{t,K+1}]]$ | |
| 8: **end for** | |

threshold $b$ (as algorithm input) we define a filter function over the state-action space $\mathcal{S} \times \mathcal{A}$:

$$\zeta(s, a; \widehat{\mu}, b) = \mathbb{1}\left(\widehat{\mu}(s, a) \geq b\right). \tag{1}$$

For simplicity we write $\zeta(s, a; \widehat{\mu}, b)$ as $\zeta(s, a)$ and define $\zeta \circ f(s, a) := \zeta(s, a)f(s, a)$. We now introduce our Pessimistic Policy Iteration (PPI) algorithm (Algorithm 1), a minor modification to vanilla FPI that constrains the Bellman backups and policy improvement step to have sufficient support on the provided data, and enforces pessimistic value estimates elsewhere. The key is to change the policy evaluation operator and only evaluate next step policy value over supported $(s, a)$ pairs, and constrain policy improvement to only optimize over supported $(s, a)$ pairs defined by $\zeta$. We define the $\zeta$-*constrained Bellman evaluation operator* $\mathcal{T}_\zeta^\pi$ as, for any $f : \mathcal{S} \times \mathcal{A} \to \mathbb{R}$,

$$(\mathcal{T}_\zeta^\pi f)(s, a) = r(s, a) + \gamma \mathbb{E}_{s'} \sum_{a' \in \mathcal{A}} \left[\pi(a'|s')\zeta \circ f(s', a')\right]. \tag{2}$$

This reduces updates that may be over-optimistic estimates in $(s', a')$'s that are not adequately covered by $\mu$ by using the most pessimistic estimate of $0$ from such pairs. There is an important difference with SPIBB [21], which still backs up $f$ values (but by estimated behavior probabilities) from such rarely visited state-action pairs: given limited data, this can lead to erroneous decisions (c.f. Fig. 2).

Given a batch dataset, we follow the common batch RL choice of least-squares residual minimization [29] and define empirical loss of $f$ given $f'$ (from last iteration) and policy $\pi$:

$$\mathcal{L}_D(f; f', \pi) := \mathbb{E}_D \left(f(s, a) - r - \gamma \sum_{a' \in \mathcal{A}} \pi(a'|s')\zeta \circ f'(s', a')\right)^2.$$

In the policy improvement step (line 7 of Algorithm 1), to ensure that the resulting policy has sufficient support, our algorithm applies the filter to the computed state-action values before performing maximization, like classification-based API [9]. We maximize using the dataset $D$ ($\mathbb{E}_D$ is a sample average) and within the policy class $\Pi$, which may not include all deterministic policies.

Analogous to PPI, we introduce Pessimistic $Q$ Iteration (PQI) (Algorithm 2) which similarly applies pessimistic estimates in the the Bellman backups where the support from data is insufficient. Define $\mathcal{T}_\zeta$ to be a $\zeta$-constrained Bellman optimality operator: for any $f : \mathcal{S} \times \mathcal{A} \to \mathbb{R}$,

$$(\mathcal{T}_\zeta f)(s, a) := r(s, a) + \gamma \mathbb{E}_{s'} \left[\max_{a'} \zeta \circ f(s', a')\right]. \tag{3}$$

Similarly, we define the empirical loss of $f$ given another function $f'$ as:

$$\mathcal{L}_D(f; f') := \mathbb{E}_D \left(f(s, a) - r - \gamma \max_{a' \in \mathcal{A}} \zeta \circ f'(s', a')\right)^2.$$

We also alter the final policy output step to only select among actions which lie in the support set (line 5 of Algorithm 2). In Figure 2 we show that our PQI and PPI can both successfully return the optimal policy when the data distribution covers the optimal policy in two illustrative examples where prior approaches struggle. The threshold $b$ is the only hyper-parameter needed for our algorithms, which trades off the risk of extrapolation with the potential benefits of Bellman backups from more state-action pairs seen in the batch of data. We discuss how practitioners can set $b$ in Section 6.

## 5 Analysis

We now provide performance bounds on the policy output by PPI: the PQI result is similar. Complete proofs for both are in the appendix. We start with some definitions and assumptions. Given $\pi$, let $\eta_h^\pi(s)$ be the marginal distribution of $s_h$ under $\pi$, that is, $\eta_h^\pi(s) := \Pr[s_h = s|\pi]$, $\eta_h^\pi(s, a) = \eta_h^\pi(s)\pi(a|s)$, and $\eta^\pi(s, a) = (1 - \gamma) \sum_{h=0}^\infty \gamma^h \eta_h^\pi(s, a)$. We make the following assumptions:

**Assumption 1** (Bounded densities)**.** *For any non-stationary policy $\pi$ and $h \geq 0$, $\eta_h^\pi(s, a) \leq U$.*

**Assumption 2** (Density estimation error). *With probability at least $1 - \delta$, $\|\widehat{\mu} - \mu\|_{TV} \leq \epsilon_\mu$.*

**Assumption 3** (Completeness under $\mathcal{T}_\zeta^\pi$). *$\forall \pi \in \Pi$, $\max_{f \in \mathcal{F}} \min_{g \in \mathcal{F}} \|g - \mathcal{T}_\zeta^\pi f\|_{2,\mu}^2 \leq \epsilon_\mathcal{F}$.*

**Assumption 4** ($\Pi$ Completeness). *$\forall f \in \mathcal{F}$, $\min_{\pi \in \Pi} \|\mathbb{E}_\pi [\zeta \circ f(s,a)] - \max_a \zeta \circ f(s,a)\|_{1,\mu} \leq \epsilon_\Pi$.*

Assumptions 3 and 4 are common but adapted to our $\zeta-$filtered operators, implying that the function class chosen is approximately complete with respect to our operator and that policy class can approximately recover the greedy policy. Assumptions 1 and 2 are novel. Assumption 2 bounds the accuracy of estimating the state–action behavior density function from limited data: $1/\sqrt{n}$ errors are standard using maximum likelihood estimation for instance [42], and the size of this error appears in the bounds. Finally Assumption 1 that the probability/density of any marginal state distribution is bounded is not very restrictive.[4] For example, this assumption holds when the density function of transitions $p(\cdot|s,a)$ and the initial state distribution are both bounded. This is always true for discrete spaces, and also holds for many distributions in continuous spaces including Gaussian distributions.

The dataset may not have sufficient samples of state–action pairs likely under the optimal policy in order to reliably and confidently return the optimal policy. Instead we hope and will show that our methods will return a policy that is close to the best policy which has sufficient support in the provided dataset. More formally, given a $\zeta$ filter over state–action pairs, we define a set of policies:

**Definition 1** ($\zeta$-constrained policy set). *Let $\Pi_C^{all}$ be the set of policies $\mathcal{S} \to \Delta(\mathcal{A})$ such that $\Pr(\zeta(s,a) = 0|\pi) \leq \epsilon_\zeta$. That is, $\mathbb{E}_{s,a \sim \eta^\pi} [\mathbb{1}(\zeta(s,a) = 0)] \leq \epsilon_\zeta$.*

$\epsilon_\zeta$ bounds the probability under a policy of escaping to state-actions with insufficient data during an episode. $\Pi_C^{all}$ adapts its size w.r.t. the filter $\zeta$ which is a function of the hyper-parameter $b$, and $\Pi_C^{all}$ does not need to be contained in $\Pi$. We now lower bound the value of the policy returned by PPI by the value of any $\widetilde{\pi} \in \Pi_C^{all}$ up to a small error, which implies a small error w.r.t. the policy with the highest value in $\Pi_C^{all}$. For ease of notation, we will denote $C = U/b$ in our results and analysis, being clear that $C$ is not assumed (as in concentrability) and is simply a function of the hyper-parameter $b$.

**Theorem 1** (Comparison with best covered policy). *When Assumptions 1 and 2 hold, given an MDP $M$, a dataset $D = \{(s,a,r,s')\}$ with $n$ samples drawn i.i.d. from $\mu \times R \times P$, and a Q-function class $\mathcal{F}$ and a policy class $\Pi$ satisfying Assumptions 3 and 4, $\widehat{\pi}_t$ from Algorithm 1 satisfies that w. p. at least $1 - 3\delta$,*

$$v_M^{\widetilde{\pi}} - v_M^{\widehat{\pi}_t} \leq \mathcal{O}\left(\frac{C\sqrt{V_{\max}^2 \ln(|\mathcal{F}||\Pi|/\delta)}}{(1-\gamma)^3\sqrt{n}}\right) + \frac{8C\sqrt{\epsilon_\mathcal{F}} + 6CV_{\max}\epsilon_\mu}{(1-\gamma)^3} + \frac{2C\epsilon_\Pi + 3\gamma^{K-1}V_{\max}}{(1-\gamma)^2} + \frac{V_{\max}\epsilon_\zeta}{1-\gamma},$$

*for any policy $\widetilde{\pi} \in \Pi_C^{all}$ and any $t \geq K$. $C = U/b$. $K$ is the number of policy evaluation iterations (inner loop) and $t$ is the number of policy improvement steps.*

*Proof sketch.* The key is to characterize the filtration effect of the conservative Bellman operators in terms of the resulting value function estimates. As an analysis tool, we construct an auxiliary MDP $M'$ by adding one additional action $a_{\text{abs}}$ in each state, leading to a zero reward absorbing state $s_{\text{abs}}$. For a *subset* of policies in $M'$, the fixed point of our conservative operator $\mathcal{T}_\zeta^\pi$ is $Q^\pi$ in $M'$. For that subset, we also have a bounded density ratio, thus we can provide the error bound in $M'$. For any $\zeta$-constrained policy, we show that it can be mapped to that subset of policies in $M'$ without a substantial loss of value, and then we finish the proof to yield a bound in $M$. □

We make a few remarks about the comparison between this result and prior related results below:

**Remark 1** (Comparison with prior API/AVI bounds). Our results match the fast rate error bound of API/AVI [7, 32, 5, 22] in their dependence on $n$. A better coefficient on $\epsilon_\mathcal{F}$ can be achieved by refining the analysis and we show this version for ease of presentation. The dependency on horizon also matches the standard API analysis and is $\mathcal{O}(1/(1-\gamma)^3)$. The dependency on $U/b$ is same as the dependency on concentrability coefficient for vanilla API analysis, but our guarantee adapts given a hyper-parameter choice instead of imposing a worst case bound over all policies.

**Remark 2** (Comparison with the theory in Kumar et al. [17]). Unlike Theorem 4.2 of Kumar et al. [17], there is no uncontrolled $f(\epsilon)$-like term. We avoid that term by being more pessimistic in our updates and by constraining the comparator class $\Pi_C^{all}$. The term $f(\epsilon)$ is not controlled by the

algorithm, because a bounded overlap in the conditional action distribution given state can potentially result in an exponentially (in horizon) small overlap in state-action joint distribution. This follows almost the same reasoning as the fact that concentrability coefficient can be exponential in horizon [5]. Hence, their algorithm and analysis combined with our comparator class $\Pi_C^{all}$ does not recover our guarantees (see also discussion of BEAR in Section 3). This is also verified conceptually by our illustrative example (b) in Figure 1b.

When the policy set $\Pi_C^{all}$ contains at least one high-value policy, Theorem 1 provides a strong guarantee. While this does not always hold, we now provide two illustrative corollaries. One when the optimal policy lies in $\Pi_C^{all}$ and another on *safe policy improvement* when $\mu$ itself lies in the set. More generally, in many situations where a concentrability-based analysis might offer nearly vacuous guarantees, we expect our theory to degrade more gracefully with the quality of the collected data.

**Corollary 1** ($\mu$ covers an optimal policy)**.** *If there exists an optimal policy $\pi^\star$ in $M$ such that* $\Pr(\mu(s,a) \leq 2b|\pi^\star) \leq \epsilon$. *Then under the conditions in Theorem 1, $\widehat{\pi}_t$ returned by Algorithm 1 satisfies that with probability at least $1 - 3\delta$, $v_M^{\widehat{\pi}_t} \geq v_M^\star - \Delta$, where $\Delta$ is the right hand side of Theorem 1 and $\epsilon_\zeta$ in $\Delta$ is $\epsilon + C\epsilon_\mu$.*

When the completeness assumptions holds without error and $\gamma^{K-1} \leq \epsilon$, the error bound reduces to

$$v_M^{\widehat{\pi}_t} \geq v_M^\star - \mathcal{O}\left( C\sqrt{\ln(|\mathcal{F}||\Pi|/\delta)/n} + C\epsilon_\mu + \epsilon \right) V_{\max}/(1-\gamma)^3$$

This is similarly tight as prior analysis assuming concentrability [19, 4]. When comparing to on-policy policy optimization [15, 34, 2, 13], the constant $C$ is akin to the density ratio with respect to an optimal policy in those works, though we pay an additional price on the size of densities. The terms regarding value function completeness can be avoided by Monte-carlo estimation in on-policy settings and there is no need for density estimation to regularize value bootstrapping either.

For PQI, the error bound takes a similar form. However the general PQI bound of Corollary 4 has an additional Bellman residual term related to $\mathcal{T}_\zeta$ and $\pi^\star$. This term arises since in value iteration, the fixed point of $\mathcal{T}_\zeta$ may no longer be the value function of the optimal policy under support, and it may not be any policy's value function.

Note that if we can find a $b$ and $\epsilon_\zeta$ such that $\mu \in \Pi_C^{all}$ given sufficient data, then Theorem 1 immediately yields a policy improvement guarantee too, analogous to the tabular guarantees known for BCQL, SPIBB and other safe policy improvement guarantees. Here, we provide the safe policy improvement guarantees in tabular settings. The details of proof are in Appendix C.5.

**Corollary 2** (Safe policy improvement – discrete state space)**.** *For finite state action spaces and $b \leq \mu_{\min}$, under the same assumptions as Theorem 1, there exist function sets $\mathcal{F}$ and $\Pi$ (specified in the proof) such that $\widehat{\pi}_t$ from Algorithm 1 satisfies that with probability at least $1 - 3\delta$,*

$$v_M^{\widehat{\pi}_t} \geq v_M^\mu - \widetilde{\mathcal{O}}\left( \frac{V_{\max}}{b(1-\gamma)^3} \left( \frac{|\mathcal{S}||\mathcal{A}|}{n} + \sqrt{\frac{|\mathcal{S}||\mathcal{A}|}{n}} \right) + \frac{\gamma^K V_{\max}}{(1-\gamma)^2} \right)$$

This corollary is comparable with the safe policy improvement result in [21] in its $\mathcal{S}, \mathcal{A}$ dependence. Note that their hyper-parameter $N_\wedge$ is analogous to $bn$ in our result. Our dependency on $1 - \gamma$ is worse, as our algorithm is not designed for the tabular setting, and matches prior results in the function approximation setting as remarked after Theorem 1.

To summarize, our analysis makes two main contributions. First, compared to prior work which provides guarantees of the performance relative to the optimal policy under concentrability, we provide similar bounds by only requiring that an optimal policy, instead of every policy, is well supported by $\mu$. Second, when an optimal policy is not well supported, our algorithms are guaranteed to output a policy whose performance is near optimal in the supported policy class.

## 6 Experimental Results

The key innovation in our algorithm uses an estimated $\zeta$ to filter backups from unsound state-action pairs. In Section 3 we showed how prior approaches without this can fail in some illustrative examples. We now experiment in two standard domains – Cartpole and Mujoco, that utilize very different $\zeta$-estimation procedures. We show several experiments where the data collected ranges

from being inadequate for batch RL to complete coverage (where even unsafe batch RL algorithms can succeed). Our algorithms return policies better than any baseline batch RL algorithm across the entire spectrum of datasets. Our algorithms need the hyperparameter $b$ to trade off conservatism of a large $b$ (where the algorithm stays at its initialization in the limit) and unfounded optimism of $b = 0$ (classical FQI/FPI). In discrete spaces, we can set $b = n_0/n$ where $n_0$ is our prior for the number of samples we need for reliable distribution estimates and $n$ is the total sample size. In continuous spaces, we can set the threshold to be a percentile of $\widehat{\mu}$, so as to filter out updates from rare outliers in the dataset. We can also run post-hoc diagnostics on the choice of $b$ by computing the average of $\zeta(s, \pi(s))$ for the resulting policy $\pi$ over the batch dataset. If this quantity is too small, we can conclude that $\zeta$ filters out too many Bellman backups and hence rerun the procedure with a lower $b$.

## 6.1 PQI in discrete spaces

We first compare PQI with AVI, BCQL [12], SPIBB [21] and behavior cloning (BC) in CartPole-v0, with a discrete state space by binning each state dimension resulting in $10^4$ states. AVI uses a vanilla Bellman operator to update the $Q$ function in each iteration. For BCQL, we fit a modified operator in their Eq (10), changing the constraint $\widehat{\mu}(a|s) > 0$ to allow a threshold $b \geq 0$. For SPIBB, we learn the $Q$ function by fitting the operator in their Eq (9) (SPIBB-DQN objective). Both SPIBB and PQI require $\widehat{\mu}(s, a)$ to construct the modified operator, but use different notions for being conservative. For all algorithms, $\widehat{\mu}(s, a)$ and $\widehat{\mu}(a|s)$ are constructed using empirical counts.

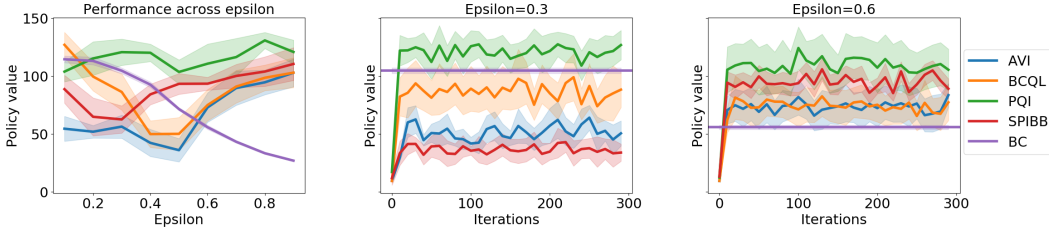

Figure 3: CartPole-v0. Left: convergent policy value across different ($\epsilon$-greedy) behavior policies. Middle and Right: learning curves when $\epsilon = 0.3, 0.6$. We allow non-zero threshold for BCQL to subsume the tabular algorithm of BEAR [17]. Shaded regions show standard deviations over 10 runs.

We collect $n = 10^4$ transitions from an epsilon-greedy policy w.r.t a near-optimal $Q$ function. We report the final policy value from different algorithms for epsilon from $0.1$ to $0.9$ in Figure 3 (left), and learning curves for $\epsilon = 0.3, 0.6$ in Figure 3 (middle, right). The results are averaged over 10 random seeds. Notice that BCQL, SPIBB, and our algorithm need a threshold of $\mu(s, a)$ or $\mu(a|s)$ as hyper-parameter. We show the results with the best threshold (in a set) of $\mu(s, a)$ for PQI and SPIBB and best threshold of $\mu(a|s)$ for BCQL. We searched over a larger set of threshold for baselines than our algorithm: {5e-4, 1e-3, 5e-3 } for PQI, {1e-4, 5e-4, 1e-3, 5e-3, 1e-2} for SPIBB and {0, 0.05, 0.1, 0.2} for BCQL. Our algorithm picks larger $b$ for smaller $\epsilon$ which matches the intuition of more conservatism when the batch dataset is not sufficiently diverse. The results show that our algorithm achieves good performance among all different $\epsilon$ values, and is always better than or close to both behavior cloning and vanilla FQI unlike other baselines.

## 6.2 PQL in continuous spaces

The core argument for PQI is that $\zeta$-filtration should focus on *state-action* $\widehat{\mu}(s, a)$ distributions rather than $\widehat{\mu}(a|s)$. To test this argument in a more complex domain, we introduce Pessimistic Q Learning (PQL) for continuous action spaces which incorporates our $\zeta$-filtration on top of the BCQ architecture [12]. PQL (like BCQ) employs an actor in continuous action space and a variational auto-encoder (VAE) to approximate $\mu(a|s)$. We use an additional VAE to fit the marginal state distribution in the dataset. Since the BCQ architecture already prevents backup from $(s, a)$ with small $\mu(a|s)$, we construct an additional filter function $\zeta(s)$ on state space by $\zeta(s) = \mathbb{1}(ELBO(s) > P_2)$ where $ELBO(s)$ is the evidence lower bound from VAE, and $P_2$ is the $2^{\text{nd}}$ percentile of ELBO values of $s$ in the whole batch. Thus the major difference between PQL and BCQ is that PQL applies additional filtration $\zeta(s')$ to the update from $s'$ in Eq (13) in [12]. Additionally, PQL is a $Q$ learning algorithm instead of actor-critic algorithm. That means in each backup step, we sample a batch of $a'$ from the VAE approximating $\mu(a|s)$ then compute max of next Q values.

| Task Name | SAC | BC | BCQ | BEAR [5] | PQL |
|---|---|---|---|---|---|
| halfcheetah-medium | -4.3 | 36.1 | **40.7** | 38.6 | 38.4 |
| hopper-medium | 0.8 | 29.0 | 54.5 | 47.6 | **75.2** |
| walker2d-medium | 0.9 | 6.6 | 53.1 | 33.2 | **68.1** |

Table 1: The final policy after 500K training steps in 3 D4RL tasks. The values are normalized with respect to the random policy (0) and expert policy (100). The results of our algorithm is averaged over 5 random seeds and the results of other algorithm are from D4RL evaluations.

We compare PQL with several state-of-the-art batch RL algorithms as well as several baselines, in a subset of tasks in the D4RL batch RL benchmark [11] (halfcheetah-medium, hopper-medium, and walker2d-medium). The data is collected by rolling out a partially trained policy using SAC for 1M steps in the corresponding MuJoCo environment. Results are given in Table 1. It shows that our algorithm is close to prior methods in the half-cheetah domain and better in the other two. As a proof-of-concept experiment, we highlight here the importance of conservative constraints on the state distribution, not only in theory but also for more practical deep RL algorithms.

# 7 Related Work

Research in batch RL focuses on deriving the best possible policy from the available data [20]. For practical settings that necessitate using function approximators, fitted value iteration [6, 33] and fitted policy iteration [19] provide an empirical foundation that has spawned many successful modern deep RL algorithms. Many prior works provide error bounds as a function of the violation of realizability and completeness assumptions such as [40]. In the online RL setting, concentrability can be side-stepped [41] but can still pose a significant challenge (e.g., the hardness of exploration in [5]). A commonly-used equivalent form of the concentrability assumption is on the discounted summation of ratios between the product of probabilities over state actions under any policy, to the data generating distribution [3, 23]. Our goal is to relax such assumptions to make the resulting algorithms more practically useful.

Several heuristics [14, 33, 12] show that algorithmic modifications help alleviate the extrapolation error or maximization bias empirically. This paper provides a simple modification that has strong theoretical guarantees even with function approximation. In contrast, the theoretical results of BEAR [17] (Theorem 4.1 and 4.2) rely on a finite state space, while the algorithm analyzed in the proof is actually the same as BCQL with a non-zero threshold whose weakness is shown in Sections 3 and 6.1. Error bound in Theorem 4.1 of [17] has an additional non-diminishing term $\alpha(\Pi)$, while non-diminishing terms in our Theorem 1 are the same as in standard analysis. Another recent work [2] highlights that sufficiently large and diverse datasets can lead to good batch RL results but we focus on the other side of the coin: a robust update rule even if we do not have a good dataset.

# 8 Discussion & Conclusion

We study a key assumption for analysis in batch value-based RL, concentrability, and provide policy iteration and $Q$ iteration algorithms with minor modifications that can be agnostic to this assumption. We remark that the other non-standard assumption about bounded density can be relaxed if we could construct the $\zeta$ filter by thresholding density ratios directly, but this results in a different filter for each policy encountered during the algorithm's operation. Being able to threshold density ratios will also allow us to assert that $\mu \in \Pi_C^{all}$ always, yielding imitation and policy improvement guarantees. We anticipate future work that develops batch RL algorithms that exploit this insight. This work can also provide some intuition for designing practical deep RL algorithms by leveraging pessimism based on the marginalized state-action distribution. Several state-of-the-art methods [17, 18] in batch RL also leverage the pessimism based on the conditional action distribution, which is composable with our proposal for state-action pessimism. Towards design more practical algorithms, estimating state action visitation distributions is an active research area (e.g. [39]) and our algorithmic framework is composable with better estimators of $\mu(s, a)$ or other uncertainty measurements.

# 9 Broader Impact

Our improvements to batch RL may improve sample efficiency of online RL and safety of off-policy RL enough to consider them in some real-world applications. However we caution that more work is needed (e.g., in closely related areas of off-policy evaluation OPE, confidence estimation, interpretability etc.) before these methods can be reliably deployed in practice. We anticipate future work in batch RL and OPE that addresses these shortcomings.

# 10 Acknowledgement

This work was supported in part by an NSF CAREER award and an ONR Young Investigator Award.

## Footnotes

[1] For detailed discussions on the unreasonableness of concentrability, please see [5, 34, 1].

[2] In their tabular algorithm BCQL the threshold is zero, but we extend it to non-zero which is also more consistent with their deep RL variant BCQ.

[3]This corresponds to a small $f(\epsilon)$ in Theorem 4.2 of Kumar et al. [17], hence in line with their theory.

[4]We need this assumption only because we filter based on $\hat{\mu}$ instead of $\eta^\pi/\mu$ during policy improvement.

[5]The performance of BEAR is from [11] that is reported before the submission. There is an improved version of BEAR code with higher performance recently.

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
