[Supplementary Material]

In Appendix A we introduce some basic definitions that are needed for our theoretical results. In Appendix B, we provide sufficient conditions for Assumption 1 that were mentioned in the main text. In Appendix C and Appendix D we prove the error bounds for PPI and PQI. In Appendix E and Appendix F we present more details of our experimental results.

# A  Definition of auxiliary MDP and policy projection

First we introduce the definition of an auxiliary MDP $M'$ based on M: each state in M has an absorbing action which leads to a self-looping absorbing state. All the other dynamics are preserved. Rewards are 0 for the absorbing action and unchanged elsewhere. More formally: The auxiliary MDP $M'$ given $M = < \mathcal{S}, \mathcal{A}, R, P, \gamma, \rho >$ is defined as $M' = < \mathcal{S}', \mathcal{A}', R', P', \gamma, \rho >$, where $\mathcal{S}' = \mathcal{S} \bigcup \{s_{\mathrm{abs}}\}$, $\mathcal{A}' = \mathcal{A} \bigcup \{a_{\mathrm{abs}}\}$. $R'$ and $P'$ are the same as $R$ and $P$ for all $(s, a) \in \mathcal{S} \times \mathcal{A}$. $R'(s, a)$ if $s = s_{\mathrm{abs}}$ or $a = a_{\mathrm{abs}}$ is a point mass on 0, and $P'(s, a)$ if $s = s_{\mathrm{abs}}$ or $a = a_{\mathrm{abs}}$ is a point mass on $s_{\mathrm{abs}}$. A data set $D$ generated from distribution $\mu$ on $M$ is also from the distribution $\mu$ on $M'$, since all distributions on $\mathcal{S} \times \mathcal{A}$ are the same between the two MDPs. This MDP is used only to perform our analysis about the error bounds on the algorithm, and is not needed at all for executing Algorithm 1 and 2. As some of the notations is actually a function of the MDP, we clarify the usage of notation w.r.t. $M/M'$ in the appendix:

1. Policy value functions $V^\pi/Q^\pi$ and Bellman operators $\mathcal{T}/\mathcal{T}^\pi$ correspond to $M'$ unless they have additional subscripts.
2. The definition of $\mathcal{F}, \Pi, \mathcal{T}_\zeta, \mathcal{T}_\zeta^\pi, \widehat{\mu}$ is independent of the change from $M$ to $M'$.
3. $\mu$ is also a distribution over $\mathcal{S}' \times \mathcal{A}'$. The definition of $\zeta$ will be extended to $\mathcal{S}' \times \mathcal{A}'$ as follow:

$$\zeta(s, a) = \begin{cases} \mathbb{1}\left(\widehat{\mu}(s, a) \geq b\right) & s \in \mathcal{S}, a \in \mathcal{A} \\ 0 & s = s_{\mathrm{abs}} \text{ or } a = a_{\mathrm{abs}} \end{cases}$$

(That means there is only one version of $\mu$ and $\zeta$ across $M$ and $M'$, instead of like we have $\mathcal{T}_{M'}^\pi$ and $\mathcal{T}_M^\pi$ for $M$ and $M'$.)

Recall the definition of semi-norm of any function of state-action pairs. For any function $g : \mathcal{S}' \times \mathcal{A}' \to \mathbb{R}, \nu \in \Delta(\mathcal{S}' \times \mathcal{A}')$, and $p \geq 1$, define the shorthand $\|g\|_{p,\nu} := (\mathbb{E}_{(s,a)\sim\nu}[|g(s,a)|^p])^{1/p}$. With some abuse of notation, later we also use this norm for $\nu \in \Delta(\mathcal{S} \times \mathcal{A})$ (specifically, $\mu$) by viewing the probability of $\nu$ on additional $(s, a)$ pairs as zero. Given a policy $\pi$, let $\eta_h^\pi(s)$ be the marginal distribution of $s_h$ under $\pi$, that is, $\eta_h^\pi(s) := \Pr[s_h = s|s_0 \sim p, \pi]$, $\eta_h^\pi(s, a) = \eta_h^\pi(s)\pi(a|s)$, and $\eta^\pi(s, a) = (1 - \gamma) \sum_{h=0}^\infty \gamma^h \eta_h^\pi(s, a)$. We also use $P(s, a)$ and $P(\nu)$ to denote the next state distribution given a state action pair or given the current state action distribution.

The norm $\|\cdot\|_{p,\nu}$ are defined over $\mathcal{S}' \times \mathcal{A}'$. Though for the input space of function $f \in \mathcal{F}$ is $\mathcal{S} \times \mathcal{A}$, the norm can still be well-defined. All of the norm would not need the value of $f(s, a)$ on $s = s_{\mathrm{abs}}$ or $a = a_{\mathrm{abs}}$, because the distribution does not cover those $(s, a)$, or the $f$ inside of the norm is multiplied by other function that is zero for those $(s, a)$.

We first formally state an obvious result about policy value in $M$ and $M'$.

**Lemma 1.** *For any policy $\pi$ that only have non-zero probability for $a \in \mathcal{A}$, $v_{M'}^\pi = v_M^\pi$.*

*Proof.* By the definition of $M'$, $P$ and $R$ are the same with $M$ over $\mathcal{S} \times \mathcal{A}$.

$$v_M^\pi = \mathbb{E}_M\left[\sum_{t=0}^h \gamma^t r_t|s_0 \sim p, \pi\right] = \mathbb{E}_{M'}\left[\sum_{t=0}^h \gamma^t r_t|s_0 \sim p, \pi\right] = v_{M'}^\pi$$

$\square$

For the readability we repeat the Definition 1 here

**Definition 1** ($\zeta$-constrained policy set ). *Let $\Pi_C^{all}$ be the set of policies $\mathcal{S} \to \Delta(\mathcal{A})$ such that $\Pr(\zeta(s, a) = 0|\pi) \leq \epsilon_\zeta$. That is*

$$(1 - \gamma) \sum_{h=0}^\infty \gamma^h \mathbb{E}_{s,a\sim\eta_h^\pi}\left[\mathbb{1}\left(\zeta(s, a) = 0\right)\right] \leq \epsilon_\zeta \tag{4}$$

41 Now we introduce another constrained policy set. Different from $\zeta$-constrained policy set which
42 we introduced in Definition 1, this policy set is on $M'$ instead of $M$ and the policy is forced to take
43 action $a_{\text{abs}}$ when $\zeta(s,a) = 0$ for all $a$. The reason we introduce this is to help us formally analyze the
44 (lower bound of) performance of the resulting policy. We essentially treat any action taken outside
45 of the support to be $a_{\text{abs}}$. Later we will define a projection to achieve that and show results about
46 how the policy value changes after projection.

47 **Definition 2** (strong $\zeta$-constrained policy set). *Let $\Pi_{SC}^{all}$ be the set of all policies $\mathcal{S}' \to \Delta(\mathcal{A}')$ such*
48 *that for $\forall (s,a)$ $\pi(a|s) > 0$ then 1) $\zeta(s,a) > 0$, or 2) $a = a_{abs}$.*

49 Notice that for $\zeta$-constrained policy set we have no requirement for $\pi$ if for any action $\zeta(s,a)$ is zero.
50 For strong $\zeta$-constrained policy set we enforce $\pi$ to take action $a_{\text{abs}}$. The second difference is $\zeta$-
51 constrained policy set requires the condition holds for $s, a$ that is reachable, which means $\eta_h^\pi(s) > 0$
52 and $\pi(a|s) > 0$. Here we require the same condition holds for any $s, a$ such that $\pi(a|s) > 0$. In
53 general, this is a stronger definition. However, we can show that for any policy in $\zeta$-constrained
54 policy set , it can be mapped to a policy in strong $\zeta$-constrained policy set , with changing value
55 bounds. Since we only need to change the behavior of policy in the state actions such that the state
56 actions that $\zeta = 0$, the value of policy will not be much different.

57 Now we define a projection that maps any policy to $\Pi_{SC}^{all}$.

58 **Definition 3** ($\zeta$-constrained policy projection). $(\Xi\pi)(a|s)$ *equals* $\zeta(s,a)\pi(a|s)$ *if $a \in \mathcal{A}$, and equals*
59 $\sum_{a' \in \mathcal{A}'} \pi(a'|s)(1 - \zeta(s,a'))$ *if $a = a_{abs}$*

60 Next we show that the projection of policy will has an equal or smaller value than the original policy.

61 **Lemma 2.** *For any policy $\pi : \mathcal{S}' \to \Delta(\mathcal{A}')$, $v_{M'}^\pi \geq v_{M'}^{\Xi(\pi)}$, and $v_{M'}^\pi = v_{M'}^{\Xi(\pi)}$ if for any $(s,a)$*
62 *reachable by $\pi$, $\zeta(s,a) = 1$.*

63 *Proof.* We drop the subscription of $M'$ in this proof for ease of notation. For any given $s$,

$$\sum_{a \in \mathcal{A}'} \pi(a|s)Q^{\Xi(\pi)}(s,a) = \sum_{a \in \mathcal{A}} \pi(a|s)Q^{\Xi(\pi)}(s,a) \qquad (Q^\pi(s, a_{\text{abs}} = 0))$$

$$\geq \sum_{a \in \mathcal{A}} \zeta(a|s)\pi(a|s)Q^{\Xi(\pi)}(s,a) \qquad (5)$$

$$= \Xi(\pi)(a_{\text{abs}}|s)Q^{\Xi(\pi)}(s, a_{\text{abs}}) + \sum_{a \in \mathcal{A}} \Xi(\pi)(a|s)Q^{\Xi(\pi)}(s,a) \quad (\text{Def of } \Xi)$$

$$= \sum_{a \in \mathcal{A}'} \Xi(\pi)(a|s)Q^{\Xi(\pi)}(s,a) \qquad (6)$$

$$= V^{\Xi(\pi)}(s) \qquad (7)$$

64 The inequality is an equality if for any $a$ s.t. $\pi(a|s) > 0$, $\zeta(s,a) = 1$. By the performance difference
65 lemma [3, Lemma 6.1]:

$$v^{\Xi(\pi)} - v^\pi = \sum_{h=0}^\infty \gamma^h \mathbb{E}_{s \sim \eta_h^\pi} \left[ V^{\Xi(\pi)}(s) - \sum_{a \in \mathcal{A}'} \pi(a|s)Q^{\Xi(\pi)}(s,a) \right] \leq 0 \qquad (8)$$

66 The inequality is an equality if for any $(s,a)$ s.t. $\eta_h^\pi(s)\pi(a|s) > 0$ for some $h$, $\zeta(s,a) = 1$.
67 In another word for any state-action reachable by $\pi$ ($\eta_h^\pi(s) > 0$ and $\pi(a|s) > 0$ for some $h$),
68 $\zeta(s,a) = 1$. $\square$

69 The following results shows for any policy $\pi$ in the $\zeta$-constrained policy set the projection will not
70 change the policy value much.

71 **Lemma 3.** *For any policy $\pi \in \Pi_C^{all}$, $v_M^\pi \leq v_{M'}^{\Xi(\pi)} + \frac{\epsilon_\zeta V_{\max}}{1-\gamma}$*

72 *Proof.* Since $\pi$ only takes action in $\mathcal{A}$, by Lemma 1, we have that $v_M^\pi = v_{M'}^\pi$. Since $\pi \in \Pi_C^{all}$, we
73 have that $\Pr(\zeta(s,a) = 0|\pi) \leq \epsilon_\zeta$, which means that:

$$(1-\gamma)\sum_{h=0}^\infty \gamma^h \mathbb{E}_{s \sim \eta_h^\pi} \left[ \mathbb{1}\left(\zeta(s,a) = 0\right) \right] \leq \epsilon_\zeta \qquad (9)$$

74 Thus:

$$v^{\Xi(\pi)} - v^{\pi} = \sum_{h=0}^{\infty} \gamma^h \mathbb{E}_{s \sim \eta_h^{\pi}} \left[ V^{\Xi(\pi)}(s) - \sum_{a \in \mathcal{A}'} \pi(a|s) Q^{\Xi(\pi)}(s,a) \right] \tag{10}$$

$$= \sum_{h=0}^{\infty} \gamma^h \mathbb{E}_{s \sim \eta_h^{\pi}} \left[ V^{\Xi(\pi)}(s) - \sum_{a \in \mathcal{A}'} \pi(a|s) \zeta(s,a) Q^{\Xi(\pi)}(s,a) \right] \tag{11}$$

$$- \sum_{h=0}^{\infty} \gamma^h \mathbb{E}_{s,a \sim \eta_h^{\pi}} \left[ \mathbb{1}\left( \zeta(s,a) = 0 \right) Q^{\Xi(\pi)}(s,a) \right] \tag{12}$$

$$\geq \sum_{h=0}^{\infty} \gamma^h \mathbb{E}_{s \sim \eta_h^{\pi}} \left[ V^{\Xi(\pi)}(s) - \sum_{a \in \mathcal{A}'} \pi(a|s) \zeta(s,a) Q^{\Xi(\pi)}(s,a) \right] \tag{13}$$

$$- V_{\max} \sum_{h=0}^{\infty} \gamma^h \mathbb{E}_{s,a \sim \eta_h^{\pi}} \left[ \mathbb{1}\left( \zeta(s,a) = 0 \right) \right] \tag{14}$$

$$\geq \sum_{h=0}^{\infty} \gamma^h \mathbb{E}_{s \sim \eta_h^{\pi}} \left[ V^{\Xi(\pi)}(s) - \sum_{a \in \mathcal{A}'} \pi(a|s) \zeta(s,a) Q^{\Xi(\pi)}(s,a) \right] - \frac{V_{\max} \epsilon_\zeta}{1 - \gamma} \tag{15}$$

$$= -\frac{V_{\max} \epsilon_\zeta}{1 - \gamma} \tag{16}$$

75 The last step follows from the first part in the proof of Lemma 2, $v_{M'}^{\pi} - v_{M'}^{\Xi(\pi)} \leq \frac{V_{\max} \epsilon_\zeta}{1-\gamma}$. □

## B  Justification of Assumption 1

77 In this section we prove a claim stated in Section 5 about the upper bound on density functions. We
78 are going to prove Assumption 1 holds under when the transition density is bounded.

79 **Lemma 4.** *Let $p(\cdot|s,a)$ be the probability density function of transition distribution: $\rho(s_0) \leq \sqrt{U} <$*
80 *$\infty, p(s_{t+1}|s_t, a_t) \leq \sqrt{U} < \infty$ and $\forall \pi(a_t|s_t, h) \leq \sqrt{U} < \infty$, for all $s_0, s_t, s_{t+1} \in \mathcal{S}$ and $a \in \mathcal{A}$.*
81 *Then in $M'$ for any non-stationary policy $\pi : \mathcal{S}' \times \mathbb{N} \to \Delta(\mathcal{A}')$ and $h \geq 0$, $\eta_h^{\pi}(s,a) \leq U$ for any*
82 *$s \in \mathcal{S}$ and $a \in \mathcal{A}$.*

83 *Proof.* We first prove that $\eta_h^{\pi}(s) \leq \sqrt{U}$ for any non-stationary policy $\pi$. For $h = 0$, $\eta_h^{\pi}(s) = \rho(s) \leq$
84 $\sqrt{U}$. For $h \geq 1$ and $s \in \mathcal{S}$:

$$\eta_h^{\pi}(s) = \int_{s_{-1} \in \mathcal{S}'} \sum_{a \in \mathcal{A}'} \eta_{h-1}^{\pi}(s_{-1}) \pi(a_{-1}|s_{-1}, h-1) p(s|s_{-1}, a_{-1}) \mathrm{d}s_{-1} \tag{17}$$

$$= \int_{s_{-1} \in \mathcal{S}} \sum_{a \in \mathcal{A}} \eta_{h-1}^{\pi}(s_{-1}) \pi(a_{-1}|s_{-1}, h-1) p(s|s_{-1}, a_{-1}) \mathrm{d}s_{-1} \tag{18}$$

$$\leq \mathbb{E}_{\eta_{h-1}^{\pi} \times \pi(h-1)} \left[ p(s|s_{-1}, a_{-1}) \right] \tag{19}$$

$$\leq \sqrt{U} \tag{20}$$

85 The first step follows from the inductive definition of $\eta_h^{\pi}(s)$. The second step follows from that $s_{\mathrm{abs}}$ is
86 absorbing state and $a_{\mathrm{abs}}$ only leads to absorbing state. The third step follows from transition density
87 $p(s|s_{-1}, a_{-1})$ is non-negative. The last step follows from that the transition density $p(s|s_{-1}, a_{-1})$
88 is the same between $M$ and $M'$ for $s, s_{-1} \in \mathcal{S}, a_{-1} \in \mathcal{A}$, and $p(s|s_{-1}, a_{-1})$ in $M$ is upper bounded
89 by $U$. Finally, the joint density function over $s$ and $a$ $\eta_h^{\pi}(s,a) = \eta_h^{\pi}(s)\pi(a|s,h)$ is bounded by $U$,
90 and we finished the proof. □

91 For the convenience of notation later we use *admissible distribution* to refer to state-action distribu-
92 tions introduced by non-stationary policy $\pi$ in $M'$. This definition is from [1]:

93 **Definition 4** (Admissible distributions). *We say a distribution or its density function $\nu \in \Delta(\mathcal{S}' \times \mathcal{A}')$*
94 *is admissible in MDP $M'$, if there exists $h \geq 0$, and a (non-stationary) policy $\pi : \mathcal{S}' \times \mathbb{N} \to \Delta(\mathcal{A}')$,*
95 *such that $\nu(s,a) = \eta_h^{\pi}(s,a)$.*

# C Proofs for Policy Iteration Guarantees

In this section we are going to prove the result of Theorem 1 using the definition of the strong $\zeta$-constrained policy set . At a high level, the proof is done in two steps. First we prove similar result to Theorem 1 for any policy in the strong $\zeta$-constrained policy set : an upper bound of $v_{M'}^{\pi} - v_{M'}^{\pi_t}$, where $\pi$ can be any policy in the strong $\zeta$-constrained policy set and $\pi_t$ is the output of the algorithm (Theorem 2, formally stated in Appendix C.4). Then we are going to show that for any policy $\pi$ in the $\zeta$-constrained policy set after a projection $\Xi$ it is in the strong $\zeta$-constrained policy set and $v_M^{\pi} \leq v_{M'}^{\Xi(\pi)} + \frac{V_{\max}\epsilon_{\zeta}}{1-\gamma}$. Then we can provide the upper bound for $v_M^{\pi} - v_M^{\pi_t}$ for any $\pi$ in $\zeta$-constrained policy set .

The proof of Theorem 2 (the $\Pi_{SC}^{all}$ version of Theorem 1, formally stated in Appendix C.4) goes as follow. First, we show the fixed point of $\mathcal{T}_{\zeta}^{\pi}$ is $Q^{\Xi(\pi)}$ for any policy $\pi$, indicating the inner loop of policy evaluation step is actually evaluating $\pi_t = \Xi(\widehat{\pi}_t)$. We prove this result formally in Lemma 6.

To bound the gap between $\pi_t$ and any policy $\widetilde{\pi}$ in the $\zeta$-constrained policy set , we use the contraction property of $\mathcal{T}_{\zeta}^{\pi}$ to recursively decompose it into a discounted summation over policy improvement gap $Q^{\pi_{t+1}} - Q^{\pi_t}$. $\widetilde{\pi}$ in the $\zeta$-constrained policy set is needed because the operator $\mathcal{T}_{\zeta}^{\pi}$ constrains the backup on the support set of $\zeta$.

Next, we bound the policy improvement gap in Lemma 12:

$$Q^{\pi_{t+1}} - Q^{\pi_t} \geq -\mathcal{O}(\|\zeta(Q^{\pi_t} - f_{t,K})\|_{1,\nu})$$

for some admissible distribution $\nu$ related to $\pi_{t+1}$. The fact that we only need to measure the error on the support set of $\zeta$ is important. It follows from the fact that both $\pi_{t+1}$ and $\pi_t$ only takes action on the support set of $\zeta$ except $a_{abs}$ which gives us a constant value. This allows us to change the measure from arbitrary distribution $\nu$ to data distribution $\mu$, *without needing concentratability*.

The rest of proof is to upper bound $\|\zeta(Q^{\pi_t} - f_{t,K})\|_{1,\nu}$ using contraction and concentration inequalities. First, $\|\zeta(Q^{\pi_t} - f_{t,K})\|_{1,\nu}$ is upper bounded by $C\|f_{t,K} - \mathcal{T}_{\zeta}^{\pi}f_{t,K}\|_{2,\mu}/(1-\gamma)$ in Lemma 9, using a standard contraction analysis technique. Notice that here we can change the measure to $\mu$ with cost $C$ to allow us to apply concentration inequality. Then Lemma 8 bounds $\|f_{t,K} - \mathcal{T}_{\zeta}^{\pi}f_{t,K}\|_{2,\mu}$ by a function of sample size $n$ and completeness error $\epsilon_{\mathcal{F}}$ using Bernstein's inequality.

While writing the proof, we will first introduce the fixed point of $\mathcal{T}_{\zeta}^{\pi}$ is $Q^{\Xi(\pi)}$ in section C.1. We prove the upper bound of the policy evaluation error $\|\zeta(Q^{\pi_t} - f_{t,K})\|_{1,\nu}$, in section C.2, and the policy improvement step in section C.3. After we proved the main theorem, we will prove when we can bound the value gap with the optimal value in Corollary 1, as we showed in the main text.

## C.1 Fixed point property

In Algorithm 1, the output policy is $\widehat{\pi}_{t+1}$. However, we will show that is actually equivalent with the following algorithm,

---
**Algorithm 3** Pessimistic Policy Iteration (PPI, repeat Algorithm 1)

---
**Input:** $D$, $\mathcal{F}$, $\Pi$, $\widehat{\mu}$, $b$
**Output:** $\widehat{\pi}_T$
Initialize $\pi_0 \in \Pi$.
**for** $t = 0$ **to** $T - 1$ **do**
    Initialize $f_{t,0} \in \mathcal{F}$
    **for** $k = 0$ **to** $K$ **do**
        // Policy Evaluation
        $f_{t,k+1} \leftarrow \arg\min_{f \in \mathcal{F}} \mathcal{L}_D(f, f_{t,k}; \pi_t)$
    **end for**
    // Policy Improvement
    $\widehat{\pi}_{t+1} \leftarrow \arg\max_{\pi \in \Pi} \mathbb{E}_D[\mathbb{E}_{\pi}[\zeta(s,a)f_{t,K}(s,a)]]$
    $\pi_{t+1} \leftarrow \Xi(\widehat{\pi}_{t+1})$
**end for**

---

128  The output policy is still $\widehat{\pi}_{t+1}$, and we know that $v^{\widehat{\pi}_{t+1}} \geq v^{\pi_{t+1}}$. So if we can lower bound $v^{\pi_{t+1}}$
129  we immediately have the lower bound on $v^{\widehat{\pi}_{t+1}}$. The only difference in algorithm is we change the
130  policy evaluation operator from $\mathcal{T}_\zeta^{\widehat{\pi}_t}$ to $\mathcal{T}_\zeta^{\pi_t}$, where $\pi_t$ is the projection of $\widehat{\pi}_t$. The following result
131  shows these two operators are actually the same. For the ease of notation, we refer to Algorithm 3
132  in our analysis.

133  **Lemma 5.** *For any policy* $\pi : \mathcal{S}' \to \Delta(\mathcal{A}')$, $\mathcal{T}_\zeta^\pi = \mathcal{T}_\zeta^{\Xi(\pi)}$.

134  *Proof.* We only need to prove for any $f$, $\mathcal{T}_\zeta^\pi f = \mathcal{T}_\zeta^{\Xi(\pi)} f$. For any $a \in \mathcal{A}$,

$$(\mathcal{T}_\zeta^\pi f)(s, a) = r(s, a) + \gamma \mathbb{E}\left[\sum_{a' \in \mathcal{A}} \pi(a'|s')\zeta(s', a')f(s', a')\right] \tag{21}$$

$$= r(s, a) + \gamma \mathbb{E}\left[\sum_{a' \in \mathcal{A}} \pi(a'|s')\zeta^2(s', a')f(s', a')\right] \tag{22}$$

$$= r(s, a) + \gamma \mathbb{E}_{s'}\left[\sum_{a' \in \mathcal{A}} \Xi(\pi_t)(a'|s')\zeta(s', a')Q^\pi(s', a')\right] \tag{23}$$

$$= (\mathcal{T}_\zeta^{\Xi(\pi)} f)(s, a) \tag{24}$$

135  For $a = a_{\text{abs}}$, $(\mathcal{T}_\zeta^\pi f)(s, a) = 0 = (\mathcal{T}_\zeta^{\Xi(\pi)} f)(s, a)$. $\qquad\square$

136  The next result is a key insight about $\mathcal{T}_\zeta^\pi$'s behavior in $M'$ that guide our analysis.

137  **Lemma 6.** *For any policy* $\pi : \mathcal{S}' \to \Delta(\mathcal{A}')$, *the fixed point solution of* $\mathcal{T}_\zeta^\pi$ *is equal to* $Q^{\Xi(\pi)}$ *on*
138  $\mathcal{S} \times \mathcal{A}$.

139  *Proof.* By definition $Q^{\Xi(\pi)}$ is the fixed point of the standard Bellman evaluation operator on $M'$:
140  $\mathcal{T}_{M'}^{\Xi(\pi)}$. So for any $(s, a) \in \mathcal{S} \times \mathcal{A}$:

$$Q^{\Xi(\pi)}(s, a) \tag{25}$$

$$= (\mathcal{T}_{M'}^{\Xi(\pi)} Q^{\Xi(\pi)})(s, a) \tag{26}$$

$$= r(s, a) + \gamma \mathbb{E}_{s'}\left[\sum_{a' \in \mathcal{A}'} \Xi(\pi)(a'|s')Q^{\Xi(\pi)}(s', a')\right] \tag{27}$$

$$= r(s, a) + \gamma \mathbb{E}_{s'}\left[\Xi(\pi)(a_{\text{abs}}|s')Q^{\Xi(\pi)}(s', a_{\text{abs}}) + \sum_{a' \in \mathcal{A}} \Xi(\pi)(a'|s')Q^{\Xi(\pi)}(s', a')\right] \tag{28}$$

$$= r(s, a) + \gamma \mathbb{E}_{s'}\left[\sum_{a' \in \mathcal{A}} \Xi(\pi)(a'|s')Q^{\Xi(\pi)}(s', a')\right] \tag{29}$$

$$= r(s, a) + \gamma \mathbb{E}_{s'}\left[\sum_{a' \in \mathcal{A}} \pi(a'|s')\zeta(s', a')Q^{\Xi(\pi)}(s', a')\right] \tag{30}$$

$$= (\mathcal{T}_\zeta^\pi Q^{\Xi(\pi)})(s, a) \tag{31}$$

141  So we proved that $Q^{\Xi(\pi)}$ is also the fixed-point solution of $\mathcal{T}_\zeta^\pi$ constrained on $\mathcal{S} \times \mathcal{A}$. $\qquad\square$

142  An obvious consequences of these two lemmas is that the fixed point solution of $\mathcal{T}_\zeta^{\pi_t} = \mathcal{T}_\zeta^{\widehat{\pi}_t}$ equals
143  $Q^{\pi_t}$ on $\mathcal{S} \times \mathcal{A}$.

## C.2 Proofs for policy evaluation step

We start with an useful result of the expected loss of the solution from empirical loss minimization, by applying a concentration inequality.

**Lemma 7.** *Given $\pi \in \Xi(\Pi)$ and Assumption 3, let $g_f^\star = \arg\min_{g\in\mathcal{F}} \|g - \mathcal{T}_\zeta^\pi f\|_{2,\mu}$, then $\|g_f^\star - \mathcal{T}_\zeta^\pi f\|_{2,\mu}^2 \leq \epsilon_{\mathcal{F}}$. The dataset $D$ is generated i.i.d. from $M$ as follows: $(s,a) \sim \mu$, $r = R(s,a)$, $s' \sim P(s,a)$. Define $\mathcal{L}_\mu(f; f', \pi) = \mathbb{E}_D\left[\mathcal{L}_D(f; f', \pi)\right]$. We have that $\forall f \in \mathcal{F}$, with probability at least $1 - \delta$,*

$$\mathcal{L}_\mu(\mathcal{T}_{\zeta,D}f; f, \pi) - \mathcal{L}_\mu(g_f^\star; f, \pi) \leq \frac{112 V_{\max}^2 \ln \frac{|\mathcal{F}||\Pi|}{\delta}}{3n} + \sqrt{\frac{64 V_{\max}^2 \ln \frac{|\mathcal{F}||\Pi|}{\delta}}{n}} \epsilon_{\mathcal{F}}$$

*where $\mathcal{T}_{\zeta,D}^\pi f = \arg\min_{g\in\mathcal{F}} \mathcal{L}_D(g; f, \pi)$.*

*Proof.* This proof is similar with the proof of Lemma 16 in [1], and we adapt it to the $\zeta$-constrained Bellman evaluation operator $\mathcal{T}_\zeta^\pi$. First, there is no difference in $\mathcal{L}_D$ and $\mathcal{L}_\mu$ between $M$ and $M'$, and the right hand side is also the same constant for $M$ and $M'$. The distribution of $D$ in $M$ and $M'$ are the same, since $\mu$ does not cover $s_{\text{abs}}$ and $a_{\text{abs}}$. So we are going to prove the inequality for $M$, and thus this bound holds for $M'$ too.

For the simplicity of notations, let $V_f^\pi(s) = \sum_{a\in\mathcal{A}} \pi(a|s)\zeta(s,a)f(s,a)$. Fix any $f, g \in \mathcal{F}$, and define

$$X(g, f, g_f^\star) := \left(g(s,a) - r - \gamma V_f^\pi(s')\right)^2 - \left(g_f^\star(s,a) - r - \gamma V_f^\pi(s')\right)^2. \tag{32}$$

Plugging each $(s, a, r, s') \in D$ into $X(g, f, g_f^\star)$, we get i.i.d. variables $X_1(g, f, g_f^\star), X_2(g, f, g_f^\star), \ldots, X_n(g, f, g_f^\star)$. It is easy to see that

$$\frac{1}{n}\sum_{i=1}^n X_i(g, f, g_f^\star) = \mathcal{L}_D(g; f, \pi) - \mathcal{L}_D(g_f^\star; f, \pi). \tag{33}$$

By the definition of $\mathcal{L}_\mu$, it is also easy to show that

$$\mathcal{L}_\mu(g; f, \pi) = \left\| g - \mathcal{T}_\zeta^\pi f \right\|_{2,\mu}^2 + \mathbb{E}_{s,a\sim\mu}\left[\mathbb{V}_{r,s'}\left(r + \gamma \sum_{a'\in\mathcal{A}} \pi(a'|s')\zeta(s',a')f(s',a')\right)\right], \tag{34}$$

where $\mathbb{V}_{r,s'}$ is the variance over conditional distribution of $r$ and $s'$ given $(s,a)$. Notice that the second part does not depends on $g$. Then

$$\mathcal{L}_\mu(g; f, \pi) - \mathcal{L}_\mu(\mathcal{T}_\zeta^\pi f; f, \pi) = \|g - \mathcal{T}_\zeta^\pi f\|_{2,\mu}^2 \tag{35}$$

Then we bound the variance of $X$:

$$\begin{aligned}
\mathbb{V}[X(g, f, g_f^\star)] &\leq \mathbb{E}[X(g, f, g_f^\star)^2] \\
&= \mathbb{E}_\mu\left[\left(\left(g(s,a) - r - \gamma V_f(s')\right)^2 - \left(g_f^\star(s,a) - r - \gamma V_f(s')\right)^2\right)^2\right] \\
&\qquad\qquad\qquad\qquad\qquad\qquad\qquad\qquad\qquad\qquad \text{(Definition of } X\text{)} \\
&= \mathbb{E}_\mu\left[\left(g(s,a) - g_f^\star(s,a)\right)^2\left(g(s,a) + g_f^\star(s,a) - 2r - 2\gamma V_f(s')\right)^2\right] \\
&\leq 4 V_{\max}^2 \mathbb{E}_\mu\left[\left(g(s,a) - g_f^\star(s,a)\right)^2\right] \\
&= 4 V_{\max}^2 \|g - g_f^\star\|_{2,\mu}^2 \\
&\leq 8 V_{\max}^2 \left(\mathbb{E}[X(g, f, g_f^\star)] + 2\epsilon_{\mathcal{F}}\right). \tag{36}
\end{aligned}$$

165     The last step holds because

$$\|g - g_f^\star\|_{2,\mu}^2$$
$$\leq 2\left(\|g - \mathcal{T}_\zeta^\pi f\|_{2,\mu}^2 + \|\mathcal{T}_\zeta^\pi f - g_f^\star\|_{2,\mu}^2\right) \qquad\qquad ((a+b)^2 \leq 2a^2 + 2b^2)$$
$$= 2\left(\|g - \mathcal{T}_\zeta^\pi f\|_{2,\mu}^2 - \|\mathcal{T}_\zeta^\pi f - g_f^\star\|_{2,\mu}^2 + 2\|\mathcal{T}_\zeta^\pi f - g_f^\star\|_{2,\mu}^2\right)$$
$$= 2\left[(\mathcal{L}_\mu(g; f, \pi) - \mathcal{L}_\mu(\mathcal{T}_\zeta^\pi f; f, \pi)) - (\mathcal{L}_\mu(g_f^\star; f, \pi) - \mathcal{L}_\mu(\mathcal{T}_\zeta^\pi f; f, \pi)) + 2\|\mathcal{T}_\zeta^\pi f - g_f^\star\|_{2,\mu}^2\right]$$
$$\text{(Equation (35))}$$
$$= 2\left[(\mathcal{L}_\mu(g; f, \pi) - \mathcal{L}_\mu(g_f^\star; f, \pi) + 2\|\mathcal{T}_\zeta^\pi f - g_f^\star\|_{2,\mu}^2\right]$$
$$= 2\left(\mathbb{E}[X(g, f, g_f^\star)] + 2\|\mathcal{T}_\zeta^\pi f - g_f^\star\|_{2,\mu}^2\right)$$
$$\leq 2(\mathbb{E}\left[X(g, f, g_f^\star)\right] + 2\epsilon_\mathcal{F})$$

166     Next, we apply (one-sided) Bernstein's inequality and union bound over all $f \in \mathcal{F}$, $g \in \mathcal{F}$, and
167     $\pi \in \Xi(\Pi)$. With probability at least $1 - \delta$, we have

$$\mathbb{E}[X(g, f, g_f^\star)] - \frac{1}{n}\sum_{i=1}^n X_i(f, f, g_f^\star) \leq \sqrt{\frac{2\mathbb{V}[X(g, f, g_f^\star)]\ln\frac{|\mathcal{F}|^2|\Pi|}{\delta}}{n}} + \frac{4V_{\max}^2 \ln\frac{|\mathcal{F}|^2|\Pi|}{\delta}}{3n}$$

$$= \sqrt{\frac{32V_{\max}^2\left(\mathbb{E}[X(g, f, g_f^\star)] + 2\epsilon_\mathcal{F}\right)\ln\frac{|\mathcal{F}||\Pi|}{\delta}}{n}} + \frac{8V_{\max}^2 \ln\frac{|\mathcal{F}||\Pi|}{\delta}}{3n}.$$
$$(37)$$

    Since $\mathcal{T}_{\zeta,D}^\pi f$ minimizes $\mathcal{L}_D(\,\cdot\,; f, \pi)$, it also minimizes $\frac{1}{n}\sum_{i=1}^n X_i(\cdot, f, g_f^\star)$. This is because the two
objectives only differ by a constant $\mathcal{L}_D(g_f^\star; f, \pi)$. Hence,

$$\frac{1}{n}\sum_{i=1}^n X_i(\mathcal{T}_{\zeta,D}^\pi f, f, g_f^\star) \leq \frac{1}{n}\sum_{i=1}^n X_i(g_f^\star, f, g_f^\star) = 0.$$

168     Then,

$$\mathbb{E}[X(\mathcal{T}_{\zeta,D}^\pi f, f, g_f^\star)] \leq 0 + \sqrt{\frac{32V_{\max}^2\left(\mathbb{E}[X(\mathcal{T}_{\zeta,D}^\pi f, f, g_f^\star)] + 2\epsilon_\mathcal{F}\right)\ln\frac{|\mathcal{F}||\Pi|}{\delta}}{n}} + \frac{8V_{\max}^2 \ln\frac{|\mathcal{F}||\Pi|}{\delta}}{3n}.$$

169     Solving for the quadratic formula,

$$\mathbb{E}[X(\mathcal{T}_{\zeta,D}^\pi f, f, g_f^\star)] \leq \sqrt{48\left(\frac{8V_{\max}^2 \ln\frac{|\mathcal{F}||\Pi|}{\delta}}{3n}\right)^2 + \frac{64V_{\max}^2 \ln\frac{|\mathcal{F}||\Pi|}{\delta}}{n}\epsilon_\mathcal{F} + \frac{56V_{\max}^2 \ln\frac{|\mathcal{F}||\Pi|}{\delta}}{3n}}$$

$$\leq \frac{(56 + 32\sqrt{3})V_{\max}^2 \ln\frac{|\mathcal{F}||\Pi|}{\delta}}{3n} + \sqrt{\frac{64V_{\max}^2 \ln\frac{|\mathcal{F}||\Pi|}{\delta}}{n}\epsilon_\mathcal{F}}$$
$$(\sqrt{a+b} \leq \sqrt{a} + \sqrt{b} \text{ and } \ln\frac{|\mathcal{F}|}{\delta} > 0)$$

$$\leq \frac{112V_{\max}^2 \ln\frac{|\mathcal{F}||\Pi|}{\delta}}{3n} + \sqrt{\frac{64V_{\max}^2 \ln\frac{|\mathcal{F}||\Pi|}{\delta}}{n}\epsilon_\mathcal{F}}$$

170     Noticing that $\mathbb{E}[X(\mathcal{T}_{\zeta,D}f, f, g_f^\star)] = \mathcal{L}_\mu(\mathcal{T}_{\zeta,D}f; f, \pi) - \mathcal{L}_\mu(g_f^\star; f, \pi)$, we complete the proof. $\qquad\square$

171     **Lemma 8** (Policy Evaluation Accuracy). *For any $t, k \geq 1$ and $\pi_t$, $f_{t,k}$ and $f_{t,k-1}$ from Algorithm*
172     *1,*

$$\left\|f_{t,k} - \mathcal{T}_\zeta^{\pi_t} f_{t,k-1}\right\|_{2,\mu}^2 \leq \epsilon_1$$

173     *where $\epsilon_1 = \frac{208V_{\max}^2 \ln\frac{|\mathcal{F}||\Pi|}{\delta}}{3n} + 2\epsilon_\mathcal{F}$.*

*Proof.*

$$\left\| f_{t,k} - \mathcal{T}_\zeta^{\pi_t} f_{t,k-1} \right\|_{2,\mu}^2$$

$$= \mathcal{L}_\mu(f_{t,k}; f_{t,k-1}, \pi_t) - \mathcal{L}_\mu(\mathcal{T}_\zeta^{\pi_t} f_{t,k-1}; f_{t,k-1}, \pi_t)$$

$$= \left( \mathcal{L}_\mu(f_{t,k}; f_{t,k-1}, \pi_t) - \mathcal{L}_\mu(g_{f_{t,k-1}}^\star; f_{t,k-1}, \pi_t) \right) - \left( \mathcal{L}_\mu(\mathcal{T}_\zeta^{\pi_t} f_{t,k-1}; f_{t,k-1}, \pi_t) - \mathcal{L}_\mu(g_{f_{t,k-1}}^\star; f_{t,k-1}, \pi_t) \right)$$

$$\leq \frac{112 V_{\max}^2 \ln \frac{|\mathcal{F}||\Pi|}{\delta}}{3n} + \sqrt{\frac{64 V_{\max}^2 \ln \frac{|\mathcal{F}||\Pi|}{\delta}}{n}} \epsilon_\mathcal{F} + \left\| g_{f_{t,k-1}}^\star - \mathcal{T}_\zeta^{\pi_t} f_{t,k-1} \right\|_{2,\mu}$$

$$\text{(Equation (35) and Lemma 7)}$$

$$\leq \frac{112 V_{\max}^2 \ln \frac{|\mathcal{F}||\Pi|}{\delta}}{3n} + \sqrt{\frac{64 V_{\max}^2 \ln \frac{|\mathcal{F}||\Pi|}{\delta}}{n}} \epsilon_\mathcal{F} + \epsilon_\mathcal{F} \qquad \text{(Definition of } g_{f_{t,k-1}}^\star \text{ and Assumption 3)}$$

$$\leq \frac{112 V_{\max}^2 \ln \frac{|\mathcal{F}||\Pi|}{\delta}}{3n} + \frac{32 V_{\max}^2 \ln \frac{|\mathcal{F}||\Pi|}{\delta}}{n} + \epsilon_\mathcal{F} + \epsilon_\mathcal{F} = \epsilon_1 \qquad (\sqrt{2ab} \leq a + b)$$

$$\square$$

From this lemma to the proof of main theorem, we are going to condition on the fact that the event in Assumption 2 holds. In the proof of the main theorem we will impose the union bound on all failures.

**Lemma 9.** *For any admissible distribution $\nu$ on $\mathcal{S}' \times \mathcal{A}'$, and any $\pi_t$ from Algorithm 1.*

$$\|\zeta(s,a)\left(f_{t,K}(s,a) - Q^{\pi_t}(s,a)\right)\|_{1,\nu} \leq \frac{C\left(\sqrt{\epsilon_1} + V_{\max}\epsilon_\mu\right)}{1 - \gamma} + \gamma^K V_{\max} \tag{38}$$

*where $\epsilon_1$ is defined in Lemma 8.*

(Although $f_{t,K}$ is only defined on $\mathcal{S} \times \mathcal{A}$, $\zeta$ is always zero for any other $(s,a)$. Thus the all values used in the proof are well-defined. Later, when it is necessary for proof, we define the value of $f_{t,K}$ outside of $\mathcal{S} \times \mathcal{A}$ to be zero. In the algorithm, we will never need to query the value of $f_{t,K}$ outside of $\mathcal{S} \times \mathcal{A}$.)

*Proof.* For any $k \geq 1$ and any distribution $\nu$ on $\mathcal{S}' \times \mathcal{A}'$:

$$\|\zeta\left(f_{t,k} - Q^{\pi_t}\right)\|_{1,\nu} \tag{39}$$

$$\leq \left\| \zeta\left(f_{t,k} - \mathcal{T}_\zeta^{\pi_t} f_{t,k-1}\right) \right\|_{1,\nu} + \left\| \zeta\left(\mathcal{T}_\zeta^{\pi_t} f_{t,k-1} - \mathcal{T}_\zeta^{\pi_t} Q^{\pi_t}\right) \right\|_{1,\nu} \tag{40}$$

$$\leq \left\| \zeta\left(f_{t,k} - \mathcal{T}_\zeta^{\pi_t} f_{t,k-1}\right) \right\|_{1,\nu} + \left\| \mathcal{T}_\zeta^{\pi_t} f_{t,k-1} - \mathcal{T}_\zeta^{\pi_t} Q^{\pi_t} \right\|_{1,\nu} \tag{41}$$

$$\leq C \left\| f_{t,k} - \mathcal{T}_\zeta^{\pi_t} f_{t,k-1} \right\|_{1,\widehat{\mu}} + \left\| \mathcal{T}_\zeta^{\pi_t} f_{t,k-1} - \mathcal{T}_\zeta^{\pi_t} Q^{\pi_t} \right\|_{1,\nu} \tag{42}$$

$$\leq C \left( \left\| f_{t,k} - \mathcal{T}_\zeta^{\pi_t} f_{t,k-1} \right\|_{1,\mu} + V_{\max}\epsilon_\mu \right) + \left\| \mathcal{T}_\zeta^{\pi_t} f_{t,k-1} - \mathcal{T}_\zeta^{\pi_t} Q^{\pi_t} \right\|_{1,\nu} \tag{43}$$

$$\leq C \left( \left\| f_{t,k} - \mathcal{T}_\zeta^{\pi_t} f_{t,k-1} \right\|_{2,\mu} + V_{\max}\epsilon_\mu \right) + \left\| \mathcal{T}_\zeta^{\pi_t} f_{t,k-1} - \mathcal{T}_\zeta^{\pi_t} Q^{\pi_t} \right\|_{1,\nu} \qquad \text{(Jensen's inequality)}$$

$$\leq C(\sqrt{\epsilon_1} + V_{\max}\epsilon_\mu) + \left\| \mathcal{T}_\zeta^{\pi_t} f_{t,k-1} - \mathcal{T}_\zeta^{\pi_t} Q^{\pi_t} \right\|_{1,\nu} \qquad \text{(Lemma 8)}$$

$$= C(\sqrt{\epsilon_1} + V_{\max}\epsilon_\mu) + \mathbb{E}_\nu \left| \gamma \mathbb{E}_{P(\nu)} \sum_{a' \in \mathcal{A}} \pi_t(a'|s')\zeta(s',a')\left(f_{t,k-1}(s',a') - Q^{\pi_t}(s',a')\right) \right| \tag{44}$$

$$= C(\sqrt{\epsilon_1} + V_{\max}\epsilon_\mu) + \mathbb{E}_\nu \left[ \gamma \mathbb{E}_{P(\nu) \times \pi_t} |\zeta(s',a')\left(f_{t,k-1}(s',a') - Q^{\pi_t}(s',a')\right)| \right] \tag{45}$$

$$\leq C(\sqrt{\epsilon_1} + V_{\max}\epsilon_\mu) + \gamma \mathbb{E}_{P(\nu) \times \pi_t} |\zeta(s',a')\left(f_{t,k-1}(s',a') - Q^{\pi_t}(s',a')\right)| \tag{46}$$

$$\leq C(\sqrt{\epsilon_1} + V_{\max}\epsilon_\mu) + \gamma \|\zeta\left(f_{t,k-1} - Q^{\pi_t}\right)\|_{1,P(\nu) \times \pi} \tag{47}$$

185  Equation (42) holds since for all $(s, a)$ s.t. $\zeta(s, a) > 0$, $\nu(s, a) \leq U \leq \frac{U}{b}\widehat{\mu}(s, a) = C\widehat{\mu}(s, a)$.
186  Equation (43) holds since the total variation distance between $\mu$ and $\widehat{\mu}$ is bounded by $\epsilon_\mu$ and the
187  Bellman error is bounded in $[-V_{\max}, V_{\max}]$. Equation (44) follows from $\pi_t \in \Pi_{SC}^{all}$. So if $\zeta(s, a) =$
188  $0$, $\pi(a|s) = 0$ for all $a \in \mathcal{A}$. Equation (45) holds since $\zeta(\cdot, a_{\text{abs}}) = 0$. The next equation follows
189  from that $\zeta = \zeta^2$.

190  Note that this holds for any admissible distribution $\nu$ on $\mathcal{S}' \times \mathcal{A}'$ and and $k$, as well as $\epsilon_1$ does not
191  depends on $k$. Repeating this for $k$ from $K$ to 1 we will have that

$$\|\zeta(s, a)\left(f_{t,K}(s, a) - Q^{\pi_t}(s, a)\right)\|_{1,\nu} \leq \frac{1 - \gamma^K}{1 - \gamma} C\left(\sqrt{\epsilon_1} + V_{\max}\epsilon_\mu\right) + \gamma^K V_{\max} \tag{48}$$

$$< \frac{C\left(\sqrt{\epsilon_1} + V_{\max}\epsilon_\mu\right)}{1 - \gamma} + \gamma^K V_{\max} \tag{49}$$

192  $\qquad\qquad\qquad\qquad\qquad\qquad\qquad\qquad\qquad\qquad\qquad\qquad\qquad\qquad\qquad\qquad\qquad\qquad\qquad\qquad$ $\square$

## C.3  Proofs for policy improvement step

**Lemma 10** (Concentration of Policy Improvement Loss). *For any $f \in \mathcal{F}$, with probability at least*
$1 - \delta$,

$$\left\|\mathbb{E}_{\widehat{\pi}_f}\left[\zeta(s, a)f(s, a)\right] - \max_{a \in \mathcal{A}}\zeta(s, a)f(s, a)\right\|_{1,\mu} \leq \epsilon_\Pi + 2V_{\max}\sqrt{\frac{\ln(|\mathcal{F}||\Pi|/\delta)}{2n}}$$

194  *where* $\widehat{\pi}_f = \arg\max_{\pi \in \Pi} \mathbb{E}_D\left[\mathbb{E}_\pi\left[\zeta(s, a)f(s, a)\right]\right]$.

195  *Proof.* Fixed $f$, define $X(s; \pi) = \max_{a \in \mathcal{A}}\zeta(s, a)f(s, a) - \mathbb{E}_\pi\left[\zeta(s, a)f(s, a)\right]$. Notice that by
196  definition $X(s; \pi)$ is always non-negative, and $\widehat{\pi}_f = \arg\max_{\pi \in \Pi} \mathbb{E}_D\left[\mathbb{E}_\pi\left[\zeta(s, a)f(s, a)\right]\right] =$
197  $\arg\min_{\pi \in \Pi} \mathbb{E}_D[X(s; \pi)]$.

Only in this proof, let $\pi_f$ be:

$$\arg\min_{\pi \in \Pi} \mathbb{E}_\mu[X(s; \pi)] = \arg\min_{\pi \in \Pi} \left\|\mathbb{E}_\pi\left[\zeta(s, a)f(s, a)\right] - \max_{a \in \mathcal{A}}\zeta(s, a)f(s, a)\right\|_{1,\mu}.$$

198  $X(s; \pi) \in [0, V_{\max}]$. By Hoeffding's inequality and union bound over all $\pi \in \Pi$, $f \in \mathcal{F}$, with
199  probability at least $1 - \delta$ for any $f$ and $\pi \neq \pi_f$,

$$\mathbb{E}_\mu[X(s; \pi)] - \mathbb{E}_D[X(s; \pi)] \leq V_{\max}\sqrt{\frac{\ln(|\mathcal{F}||\Pi|/\delta)}{2n}} \tag{50}$$

200  for $\pi = \pi_f$

$$\mathbb{E}_D[X(s; \pi)] - \mathbb{E}_\mu[X(s; \pi)] \leq V_{\max}\sqrt{\frac{\ln(|\mathcal{F}||\Pi|/\delta)}{2n}} \tag{51}$$

201  If $\widehat{\pi}_f = \pi_f$, then $\mathbb{E}_\mu[X(s; \widehat{\pi}_f)] \leq \epsilon_\Pi$. Otherwise,

$$\mathbb{E}_\mu[X(s; \widehat{\pi}_f)] \tag{52}$$

$$\leq \mathbb{E}_D[X(s; \widehat{\pi}_f)] + V_{\max}\sqrt{\frac{\ln(|\mathcal{F}||\Pi|/\delta)}{2n}} \tag{53}$$

$$\leq \mathbb{E}_D[X(s; \pi_f)] + V_{\max}\sqrt{\frac{\ln(|\mathcal{F}||\Pi|/\delta)}{2n}} \tag{54}$$

$$\leq \mathbb{E}_\mu[X(s; \pi_f)] + 2V_{\max}\sqrt{\frac{\ln(|\mathcal{F}||\Pi|/\delta)}{2n}} \tag{55}$$

$$= \min_{\pi \in \Pi} \left\|\mathbb{E}_{\widehat{\pi}}\left[\zeta(s, a)f(s, a)\right] - \max_{a \in \mathcal{A}}\zeta(s, a)f(s, a)\right\|_{1,\mu} + 2V_{\max}\sqrt{\frac{\ln(|\mathcal{F}||\Pi|/\delta)}{2n}} \tag{56}$$

$$= \epsilon_\Pi + 2V_{\max}\sqrt{\frac{\ln(|\mathcal{F}||\Pi|/\delta)}{2n}} \tag{57}$$

202  $\qquad\qquad\qquad\qquad\qquad\qquad\qquad\qquad\qquad\qquad\qquad\qquad\qquad\qquad\qquad\qquad\qquad\qquad\qquad\qquad$ $\square$

For the following proof until the main theorem, we are going to condition on the fact that the high probability bound in the lemma above holds, and impose an union bound in the proof of main theorem.

**Lemma 11.** *For any admissible distribution $\nu$ on $\mathcal{S}'$, any policy $\pi : \mathcal{S}' \to \Delta(\mathcal{A}')$,*

$$\mathbb{E}_\nu \left[ \mathbb{E}_{\pi_{t+1}} \left[ \zeta(s,a) f_{t,K}(s,a) \right] - \mathbb{E}_\pi \left[ \zeta(s,a) f_{t,K}(s,a) \right] \right] \geq$$
$$-C \left( \epsilon_\Pi + V_{\max} \epsilon_\mu + 2 V_{\max} \sqrt{\frac{\ln(|\mathcal{F}||\Pi|/\delta)}{2n}} \right)$$

*Proof.* Recall that $\pi_{t+1} = \Xi(\widehat{\pi}_{t+1})$. So $\pi_{t+1}(a|s) = \widehat{\pi}_{t+1}(a|s)$ for all $a$ such that $\zeta(s,a) = 1$. Then

$$\mathbb{E}_{\pi_{t+1}} \left[ \zeta(s,a) f_{t,K}(s,a) \right] = \mathbb{E}_{\widehat{\pi}_{t+1}} \left[ \zeta(s,a) f_{t,K}(s,a) \right]$$
$$\mathbb{E}_\nu \left[ \mathbb{E}_{\pi_{t+1}} \left[ \zeta(s,a) f_{t,K}(s,a) \right] \right] = \mathbb{E}_\nu \left[ \mathbb{E}_{\widehat{\pi}_{t+1}} \left[ \zeta(s,a) f_{t,K}(s,a) \right] \right]$$

$$\mathbb{E}_\nu \left[ \mathbb{E}_{\pi_{t+1}} \left[ \zeta(s,a) f_{t,K}(s,a) \right] - \mathbb{E}_\pi \left[ \zeta(s,a) f_{t,K}(s,a) \right] \right] \tag{58}$$
$$= \mathbb{E}_\nu \left[ \mathbb{E}_{\widehat{\pi}_{t+1}} \left[ \zeta(s,a) f_{t,K}(s,a) \right] - \mathbb{E}_\pi \left[ \zeta(s,a) f_{t,K}(s,a) \right] \right] \tag{59}$$
$$= \mathbb{E}_\nu \left[ \mathbb{E}_{\widehat{\pi}_{t+1}} \left[ \zeta(s,a) f_{t,K}(s,a) \right] - \max_{a \in \mathcal{A}} \zeta(s,a) f_{t,K}(s,a) + \max_{a \in \mathcal{A}} \zeta(s,a) f_{t,K}(s,a) - \mathbb{E}_\pi \left[ \zeta(s,a) f_{t,K}(s,a) \right] \right] \tag{60}$$
$$\geq \mathbb{E}_\nu \left[ \mathbb{E}_{\widehat{\pi}_{t+1}} \left[ \zeta(s,a) f_{t,K}(s,a) \right] - \max_{a \in \mathcal{A}} \zeta(s,a) f_{t,K}(s,a) \right] \tag{61}$$
$$\geq -\mathbb{E}_\nu \left| \mathbb{E}_{\widehat{\pi}_{t+1}} \left[ \zeta(s,a) f_{t,K}(s,a) \right] - \max_{a \in \mathcal{A}} \zeta(s,a) f_{t,K}(s,a) \right| \tag{62}$$
$$= -\left\| \mathbb{E}_{\widehat{\pi}_{t+1}} \left[ \zeta(s,a) f_{t,K}(s,a) \right] - \max_{a \in \mathcal{A}} \zeta(s,a) f_{t,K}(s,a) \right\|_{1,\nu} \tag{63}$$
$$\geq -C \left\| \mathbb{E}_{\widehat{\pi}_{t+1}} \left[ \zeta(s,a) f_{t,K}(s,a) \right] - \max_{a \in \mathcal{A}} \zeta(s,a) f_{t,K}(s,a) \right\|_{1,\widehat{\mu}} \tag{64}$$

The last step follows from that $\zeta(s,a) = 1 \Rightarrow \widehat{\mu}(s,a) \geq b \Rightarrow \widehat{\mu}(s) \geq b \Rightarrow -\nu(s) \geq -U \geq -C\widehat{\mu}(s)$, and for all other $(s,a)$ the term inside of norm is zero. Since the total variation distance between $\widehat{\mu}$ and $\mu$ is bounded by $\epsilon_\mu$

$$\left\| \mathbb{E}_{\widehat{\pi}_{t+1}} \left[ \zeta(s,a) f_{t,K}(s,a) \right] - \max_{a \in \mathcal{A}} \zeta(s,a) f_{t,K}(s,a) \right\|_{1,\widehat{\mu}} \tag{65}$$
$$\leq \left\| \mathbb{E}_{\widehat{\pi}_{t+1}} \left[ \zeta(s,a) f_{t,K}(s,a) \right] - \max_{a \in \mathcal{A}} \zeta(s,a) f_{t,K}(s,a) \right\|_{1,\mu} + V_{\max} \epsilon_\mu \tag{66}$$

By Lemma 10:

$$\left\| \mathbb{E}_{\widehat{\pi}_{t+1}} \left[ \zeta(s,a) f_{t,K}(s,a) \right] - \max_{a \in \mathcal{A}} \zeta(s,a) f_{t,K}(s,a) \right\|_{1,\mu} \leq \epsilon_\Pi + 2 V_{\max} \sqrt{\frac{\ln(|\mathcal{F}||\Pi|/\delta)}{2n}} \tag{67}$$

Then we finished the proof by plug this into the last equation. $\square$

**Lemma 12.** *For any $(s,a) \in \mathcal{S}' \times \mathcal{A}'$, and any $\pi_t$, $\pi_{t+1}$ in Algorithm 1,*

$$Q^{\pi_{t+1}}(s,a) - Q^{\pi_t}(s,a) \geq -\frac{2C\sqrt{\epsilon_1} + 3V_{\max}C\epsilon_\mu}{(1-\gamma)^2} - \frac{\epsilon_2 + 2\gamma^K V_{\max}}{1-\gamma} \tag{68}$$

*where $\epsilon_1$ is defined in Lemma 8, $\epsilon_2 = C \left( \epsilon_\Pi + 2 V_{\max} \sqrt{\frac{\ln(|\mathcal{F}||\Pi|/\delta)}{2n}} \right)$.*

215 *Proof.* For any $s'$, only in this proof, let $\eta_h^{\pi_{t+1}}$ be the state distribution on the $h$th step from initial
216 state $s'$ following $\pi_{t+1}$. By applying performance difference lemma [3],

$$V^{\pi_{t+1}}(s') - V^{\pi_t}(s') \tag{69}$$

$$= \sum_{h=1}^{\infty} \gamma^{h-1} \mathbb{E}_{z \sim \eta_h^{\pi_{t+1}}} \left[ \sum_{a \in \mathcal{A}'} \left( \pi_{t+1}(a|z) Q^{\pi_t}(z,a) - \pi_t(a|z) Q^{\pi_t}(z,a) \right) \right] \tag{70}$$

$$= \sum_{h=1}^{\infty} \gamma^{h-1} \mathbb{E}_{z \sim \eta_h^{\pi_{t+1}}} \left[ \sum_{a \in \mathcal{A}'} (1 - \zeta(z,a)) \left( \pi_{t+1}(a|z) Q^{\pi_t}(z,a) - \pi_t(a|z) Q^{\pi_t}(z,a) \right) \right. \tag{71}$$

$$\left. + \sum_{a \in \mathcal{A}'} \zeta(z,a) \left( \pi_{t+1}(a|z) Q^{\pi_t}(z,a) - \pi_t(a|z) Q^{\pi_t}(z,a) \right) \right] \tag{72}$$

217 Because $\pi_t, \pi_{t+1} \in \Pi_{SC}^{all}$, $\zeta(z,a) = 0$ means either $\pi_t(a|z) = \pi_{t+1}(a|z) = 0$ or $a = a_{\text{abs}}$. So the
218 first term is zero. Then:

$$V^{\pi_{t+1}}(s') - V^{\pi_t}(s') \tag{73}$$

$$= \sum_{h=1}^{\infty} \gamma^{h-1} \mathbb{E}_{z \sim \eta_h^{\pi_{t+1}}} \left[ \sum_{a \in \mathcal{A}'} \zeta(z,a) \left( \pi_{t+1}(a|z) Q^{\pi_t}(z,a) - \pi_t(a|z) Q^{\pi_t}(z,a) \right) \right] \tag{74}$$

$$= \sum_{h=1}^{\infty} \gamma^{h-1} \mathbb{E}_{z \sim \eta_h^{\pi_{t+1}}} \left[ \sum_{a \in \mathcal{A}} \zeta(z,a) \left( \pi_{t+1}(a|z) Q^{\pi_t}(z,a) - \pi_t(a|z) Q^{\pi_t}(z,a) \right) \right] \tag{75}$$

$$= \sum_{h=1}^{\infty} \gamma^{h-1} \mathbb{E}_{z \sim \eta_h^{\pi_{t+1}}} \left[ \sum_{a \in \mathcal{A}} \zeta(z,a) \left( \pi_{t+1}(a|z) Q^{\pi_t}(z,a) - \pi_{t+1}(a|z) f_{t,K}(z,a) \right) \right. \tag{76}$$

$$+ \sum_{a \in \mathcal{A}} \zeta(z,a) \left( \pi_{t+1}(a|z) f_{t,K}(z,a) - \pi_t(a|z) f_{t,K}(z,a) \right) \tag{77}$$

$$\left. + \sum_{a \in \mathcal{A}} \zeta(z,a) \left( \pi_t(a|z) f_{t,K}(z,a) - \pi_t(a|z) Q^{\pi_t}(z,a) \right) \right] \tag{78}$$

219 Equation 75 follows from $Q^\pi(s, a_{\text{abs}}) = 0$ for any $\pi$ and $s$. By Lemma 11, for any $h$,

$$\mathbb{E}_{z \sim \eta_h^{\pi_{t+1}}} \left[ \sum_{a \in \mathcal{A}} \zeta(z,a) \left( \pi_{t+1}(a|z) f_{t,K}(z,a) - \pi_t(a|z) f_{t,K}(z,a) \right) \right] \tag{79}$$

$$= \mathbb{E}_{z \sim \eta_h^{\pi_{t+1}}} \left[ \mathbb{E}_{\pi_{t+1}} \left[ \zeta(s,a) f_{t,K}(s,a) \right] - \mathbb{E}_{\pi_t} \left[ \zeta(s,a) f_{t,K}(s,a) \right] \right] \geq -\epsilon_2 - CV_{\max} \epsilon_\mu \tag{80}$$

220 Then

$$V^{\pi_{t+1}}(s') - V^{\pi_t}(s') \tag{81}$$

$$\geq \sum_{h=1}^{\infty} \gamma^{h-1} \mathbb{E}_{z \sim \eta_h^{\pi_{t+1}}} \left[ \sum_{a \in \mathcal{A}} \zeta(z,a) \left( \pi_{t+1}(a|z) Q^{\pi_t}(z,a) - \pi_{t+1}(a|z) f_{t,K}(z,a) \right) \right. \tag{82}$$

$$\left. + \sum_{a \in \mathcal{A}} \zeta(z,a) \left( \pi_t(a|z) f_{t,K}(z,a) - \pi_t(a|z) Q^{\pi_t}(z,a) \right) \right] - \frac{\epsilon_2 + CV_{\max}\epsilon_\mu}{1-\gamma} \tag{83}$$

$$\geq -\sum_{h=1}^{\infty} \gamma^{h-1} \left( \| \zeta(z,a)(Q^{\pi_t}(z,a) - f_{t,K}(z,a)) \|_{1, \eta_h^{\pi_{t+1}}} \right. \tag{84}$$

$$\left. + \| \zeta(z,a)(Q^{\pi_t}(z,a) - f_{t,K}(z,a)) \|_{1, \eta_h^{\pi_{t+1}} \times \pi_t} \right) - \frac{\epsilon_2 + CV_{\max}\epsilon_\mu}{1-\gamma} \tag{85}$$

$$\geq -\sum_{h=1}^{\infty} \gamma^{h-1} \left( \| \zeta(z,a)(Q^{\pi_t}(z,a) - f_{t,K}(z,a)) \|_{2, \eta_h^{\pi_{t+1}}} \right. \tag{86}$$

$$\left. + \| \zeta(z,a)(Q^{\pi_t}(z,a) - f_{t,K}(z,a)) \|_{2, \eta_h^{\pi_{t+1}} \times \pi_t} \right) - \frac{\epsilon_2 + CV_{\max}\epsilon_\mu}{1-\gamma} \tag{87}$$

$$\geq \frac{-2C\left(\sqrt{\epsilon_1} + V_{\max}\epsilon_\mu\right)}{(1-\gamma)^2} - \frac{2\gamma^K V_{\max}}{1-\gamma} - \frac{\epsilon_2 + CV_{\max}\epsilon_\mu}{1-\gamma} \tag{Lemma 9}$$

221 Equation 87 follows from Jensen's inequality. Since this holds for any $s'$, we proved that for any
222 $(s,a)$,

$$[Q^{\pi_{t+1}}(s,a) - Q^{\pi_t}(s,a)] \tag{88}$$

$$= \gamma \mathbb{E}_{s'} \left[ V^{\pi_{t+1}}(s') - V^{\pi_t}(s') \right] \tag{89}$$

$$\geq \frac{-2C\left(\sqrt{\epsilon_1} + V_{\max}\epsilon_\mu\right)}{(1-\gamma)^2} - \frac{2\gamma^K V_{\max}}{1-\gamma} - \frac{\epsilon_2 + CV_{\max}\epsilon_\mu}{1-\gamma} \tag{90}$$

$$\geq -\frac{2C\sqrt{\epsilon_1} + 3CV_{\max}\epsilon_\mu}{(1-\gamma)^2} - \frac{2\gamma^K V_{\max}}{1-\gamma} - \frac{\epsilon_2}{1-\gamma} \tag{91}$$

223 $\qquad\qquad\qquad\qquad\qquad\qquad\qquad\qquad\qquad\qquad\qquad\qquad\qquad\qquad\qquad\qquad\qquad\qquad\quad \square$

## 224 C.4 Proof of main theorems

225 **Theorem 2.** *Given an MDP $M = <\mathcal{S}, \mathcal{A}, R, P, \gamma, p>$, a dataset $D = \{(s,a,r,s')\}$ with $n$ samples*
226 *that is draw i.i.d. from $\mu \times R \times P$, and a finite Q-function classes $\mathcal{F}$ and a finite policy class $\Pi$*
227 *satisfying Assumption 3 and 4, $\pi_t = \Xi(\widehat{\pi}_t)$ from Algorithm 1 satisfies that with probability at least*
228 $1 - 3\delta$,

$$v^{\widetilde{\pi}} - v^{\pi_t} \leq \frac{4C}{(1-\gamma)^3} \left( \sqrt{\frac{419 V_{\max}^2 \ln \frac{|\mathcal{F}||\Pi|}{\delta}}{3n}} + 2\sqrt{\epsilon_{\mathcal{F}}} \right) + \frac{6CV_{\max}\epsilon_\mu}{(1-\gamma)^3} + \frac{2C\epsilon_\Pi + 3\gamma^{K-1} V_{\max}}{(1-\gamma)^2}$$

229 *for any policy $\widetilde{\pi} \in \Pi_{SC}^{all}$.*

230 *Proof.* For simplicity of the notation, let $\epsilon_1 = \frac{208 V_{\max}^2 \ln \frac{|\mathcal{F}||\Pi|}{\delta}}{3n} + 2\epsilon_{\mathcal{F}}$, $\epsilon_2 =$
231 $C\left( \epsilon_\Pi + 2V_{\max}\sqrt{\frac{\ln(|\mathcal{F}||\Pi|/\delta)}{2n}} \right)$ and $\epsilon_3 = \frac{2C\sqrt{\epsilon_1} + 3V_{\max}C\epsilon_\mu}{(1-\gamma)^2} + \frac{\epsilon_2 + 2\gamma^K V_{\max}}{1-\gamma}$. We start by proving
232 a stronger result. For any $\widetilde{\pi} \in \Pi_{SC}^{all}$, we will upper bound $\mathbb{E}_\nu[V^{\widetilde{\pi}} - V^{\pi_t}]$ for any admissible

233 distribution $\nu$ over $\mathcal{S}'$ which will naturally be an upper bound for $v^{\widetilde{\pi}} - v^{\pi_t}$

$$\mathbb{E}_\nu[V^{\widetilde{\pi}} - V^{\pi_{t+1}}]$$

$$= \mathbb{E}_\nu \left[ V^{\widetilde{\pi}}(s) - \sum_{a \in \mathcal{A}'} \pi_{t+1}(a|s)Q^{\pi_t}(s,a) + \sum_{a \in \mathcal{A}'} \pi_{t+1}(a|s)Q^{\pi_t}(s,a) - V^{\pi_{t+1}}(s) \right]$$

$$= \mathbb{E}_\nu \left[ V^{\widetilde{\pi}}(s) - \sum_{a \in \mathcal{A}'} \pi_{t+1}(a|s)Q^{\pi_t}(s,a) + \sum_{a \in \mathcal{A}'} \pi_{t+1}(a|s) \left( Q^{\pi_t}(s,a) - Q^{\pi_{t+1}}(s,a) \right) \right]$$

$$\leq \mathbb{E}_\nu \sum_{a \in \mathcal{A}'} \left[ \widetilde{\pi}(a|s)Q^{\widetilde{\pi}}(s,a) - \pi_{t+1}(a|s)Q^{\pi_t}(s,a) \right] + \epsilon_3 \qquad \text{(Lemma 12)}$$

$$= \mathbb{E}_\nu \sum_{a \in \mathcal{A}'} \zeta(s,a)[\widetilde{\pi}(a|s)Q^{\widetilde{\pi}}(s,a) - \pi_{t+1}(a|s)Q^{\pi_t}(s,a)] + \epsilon_3$$

$$= \mathbb{E}_\nu \left[ \mathbb{E}_{\widetilde{\pi}} \left[ \zeta(s,a)Q^{\widetilde{\pi}}(s,a) \right] - \mathbb{E}_{\pi_{t+1}} \left[ \zeta(s,a)f_t(s,a) \right] \right.$$

$$\left. + \mathbb{E}_{\pi_{t+1}} \left[ \zeta(s,a)f_t(s,a) \right] - \mathbb{E}_{\pi_{t+1}} \left[ \zeta(s,a)Q^{\pi_{t+1}}(s,a) \right] \right] + \epsilon_3$$

$$\leq \mathbb{E}_\nu \left[ \mathbb{E}_{\widetilde{\pi}} \left[ \zeta(s,a)Q^{\widetilde{\pi}}(s,a) \right] - \mathbb{E}_{\pi_{t+1}} \left[ \zeta(s,a)f_t(s,a) \right] \right]$$

$$+ \| \zeta(z,a)(Q^{\pi_t}(z,a) - f_t(z,a)) \|_{1, \nu \times \pi_{t+1}} + \epsilon_3$$

$$\leq \mathbb{E}_\nu \left[ \mathbb{E}_{\widetilde{\pi}} \left[ \zeta(s,a)Q^{\widetilde{\pi}}(s,a) \right] - \mathbb{E}_{\pi_{t+1}} \left[ \zeta(s,a)f_t(s,a) \right] \right] + \frac{C\sqrt{\epsilon_1} + CV_{\max}\epsilon_\mu}{1 - \gamma} + \gamma^K V_{\max} + \epsilon_3$$

$$\text{(Lemma 9)}$$

$$\leq \mathbb{E}_\nu \left[ \mathbb{E}_{\widetilde{\pi}} \left[ \zeta(s,a)Q^{\widetilde{\pi}}(s,a) \right] - \mathbb{E}_{\widetilde{\pi}} \left[ \zeta(s,a)f_t(s,a) \right] \right] + \epsilon_2 + CV_{\max}\epsilon_\mu + \frac{C\sqrt{\epsilon_1} + CV_{\max}\epsilon_\mu}{1 - \gamma} + \gamma^K V_{\max} + \epsilon_3$$

$$\text{(Lemma 11)}$$

$$\leq \mathbb{E}_\nu \left[ \mathbb{E}_{\widetilde{\pi}} \left[ \zeta(s,a)Q^{\widetilde{\pi}}(s,a) \right] - \mathbb{E}_{\widetilde{\pi}} \left[ \zeta(s,a)Q^{\pi_t}(s,a) \right] \right] + \epsilon_2 + \frac{2C\sqrt{\epsilon_1} + 3CV_{\max}\epsilon_\mu}{1 - \gamma} + 2\gamma^K V_{\max} + \epsilon_3$$

$$\text{(Lemma 9)}$$

$$= \mathbb{E}_{\nu \times \widetilde{\pi}} \left[ \zeta(s,a)Q^{\widetilde{\pi}}(s,a) - \zeta(s,a)Q^{\pi_t}(s,a) \right] + \epsilon_2 + \frac{2C\sqrt{\epsilon_1} + 3CV_{\max}\epsilon_\mu}{1 - \gamma} + 2\gamma^K V_{\max} + \epsilon_3$$

$$= \mathbb{E}_{\nu \times \widetilde{\pi}} \left[ Q^{\widetilde{\pi}}(s,a) - Q^{\pi_t}(s,a) \right] + \epsilon_2 + \frac{2C\sqrt{\epsilon_1} + 3CV_{\max}\epsilon_\mu}{1 - \gamma} + 2\gamma^K V_{\max} + \epsilon_3 \qquad (\pi_t \in \Pi_{SC}^{all})$$

$$\leq \gamma \mathbb{E}_{P(\nu \times \widetilde{\pi})}[V^{\widetilde{\pi}} - V^{\pi_t}] + \epsilon_2 + \frac{2C\sqrt{\epsilon_1} + 3CV_{\max}\epsilon_\mu}{1 - \gamma} + 2\gamma^K V_{\max} + \epsilon_3$$

234 The second to last step follows from $\pi_t \in \Pi_{SC}^{all}$: for all $s, a$ such that $\widetilde{\pi}(a|s) > 0$, either $\zeta(s,a) = 1$,
235 or $a = a_{\text{abs}}$. The later two indicate that $Q^{\pi_t}(s,a) = Q^{\widetilde{\pi}}(s,a) = 0$. So for all $s, a$ such that
236 $\widetilde{\pi}(a|s) > 0$, $Q^{\widetilde{\pi}}(s,a) = \zeta(s,a)Q^{\widetilde{\pi}}(s,a)$ and $Q^{\pi_t}(s,a) = \zeta(s,a)Q^{\pi_t}(s,a)$. Now we proved

$$\mathbb{E}_\nu[V^{\widetilde{\pi}} - V^{\pi_{t+1}}] \leq \gamma \mathbb{E}_{P(\nu \times \widetilde{\pi})}[V^{\widetilde{\pi}} - V^{\pi_t}] + \epsilon_2 + \epsilon_3 + \frac{2C\sqrt{\epsilon_1} + 3CV_{\max}\epsilon_\mu}{1 - \gamma} + 2\gamma^K V_{\max} \quad (92)$$

holds for any distribution $\nu$. The error terms do not depend on $t$ and this holds for any $t$. We can repeatedly apply this for all $0 < t' \leq t$. Assuming $t \geq K$ this will give us :

$$
\mathbb{E}_\nu[V^{\widetilde{\pi}} - V^{\pi_{t+1}}]
$$

$$
\leq \frac{1-\gamma^t}{1-\gamma} \left( \epsilon_2 + \epsilon_3 + \frac{2C\sqrt{\epsilon_1} + 3CV_{\max}\epsilon_\mu}{1-\gamma} + 2\gamma^K V_{\max} \right) + \gamma^t V_{\max}
$$

$$
\leq \frac{\epsilon_2}{1-\gamma} + \frac{\epsilon_3}{1-\gamma} + \frac{2C\sqrt{\epsilon_1}}{(1-\gamma)^2} + \frac{3CV_{\max}\epsilon_\mu}{(1-\gamma)^2} + \frac{3\gamma^K V_{\max}}{1-\gamma}
$$

$$
\leq \frac{2\epsilon_2}{(1-\gamma)^2} + \frac{4C\sqrt{\epsilon_1}}{(1-\gamma)^3} + \frac{6CV_{\max}\epsilon_\mu}{(1-\gamma)^3} + \frac{3\gamma^{K-1} V_{\max}}{(1-\gamma)^2}
$$

$$
\leq \frac{2C\epsilon_\Pi}{(1-\gamma)^2} + \frac{4C}{(1-\gamma)^2}\sqrt{\frac{V_{\max}^2 \ln(|\mathcal{F}||\Pi|/\delta)}{2n}} + \frac{4C\sqrt{\epsilon_1}}{(1-\gamma)^3} + \frac{6CV_{\max}\epsilon_\mu}{(1-\gamma)^3} + \frac{3\gamma^{K-1} V_{\max}}{(1-\gamma)^2}
$$

$$
\leq \frac{2C\epsilon_\Pi}{(1-\gamma)^2} + \frac{4C}{(1-\gamma)^3} \left( \sqrt{\frac{V_{\max}^2 \ln(|\mathcal{F}||\Pi|/\delta)}{2n}} + \sqrt{\frac{208 V_{\max}^2 \ln(|\mathcal{F}||\Pi|/\delta)}{3n} + 2\epsilon_\mathcal{F}} \right)
$$

$$
+ \frac{6CV_{\max}\epsilon_\mu}{(1-\gamma)^3} + \frac{3\gamma^{K-1} V_{\max}}{(1-\gamma)^2}
$$

$$
\leq \frac{2C\epsilon_\Pi}{(1-\gamma)^2} + \frac{4C}{(1-\gamma)^3} \left( \sqrt{\frac{V_{\max}^2 \ln(|\mathcal{F}||\Pi|/\delta)}{2n}} + \sqrt{\frac{208 V_{\max}^2 \ln(|\mathcal{F}||\Pi|/\delta)}{3n}} + \sqrt{2\epsilon_\mathcal{F}} \right)
$$

$$
+ \frac{6CV_{\max}\epsilon_\mu}{(1-\gamma)^3} + \frac{3\gamma^{K-1} V_{\max}}{(1-\gamma)^2}
$$

$$
\leq \frac{2C\epsilon_\Pi}{(1-\gamma)^2} + \frac{4C}{(1-\gamma)^3} \left( \sqrt{\frac{419 V_{\max}^2 \ln(|\mathcal{F}||\Pi|/\delta)}{3n}} + \sqrt{2\epsilon_\mathcal{F}} \right) + \frac{6CV_{\max}\epsilon_\mu}{(1-\gamma)^3} + \frac{3\gamma^{K-1} V_{\max}}{(1-\gamma)^2}
$$

The last step follows from that $a + b \leq \sqrt{2(a^2 + b^2)}$. The error bound is finished by simplifying the expression. The failure probability $3\delta$ is from the union bound of probability $\delta$ on which Assumption 2 fails, probability $\delta$ on which Lemma 7 fails, and the probability $\delta$ on which Lemma 10 fails. $\quad\square$

Now we are going to use the fact that there is an almost no-value-loss projection from the $\zeta$-constrained policy set to the strong $\zeta$-constrained policy set in order to prove an error bound w.r.t any $\widetilde{\pi} \in \Pi_C^{all}$.

**Theorem 1.** *Given an MDP $M = <\mathcal{S}, \mathcal{A}, R, P, \gamma, p>$, a dataset $D = \{(s,a,r,s')\}$ with $n$ samples that is draw i.i.d. from $\mu \times R \times P$, and a finite Q-function classes $\mathcal{F}$ and a finite policy class $\Pi$ satisfying Assumption 3 and 4, $\widehat{\pi}_t$ from Algorithm 1 satisfies that with probability at least $1 - 3\delta$,*

$$
v_M^{\widetilde{\pi}} - v_M^{\widehat{\pi}_t} \leq \frac{4C}{(1-\gamma)^3} \left( \sqrt{\frac{419 V_{\max}^2 \ln \frac{|\mathcal{F}||\Pi|}{\delta}}{3n}} + 2\sqrt{\epsilon_\mathcal{F}} \right) + \frac{6CV_{\max}\epsilon_\mu}{(1-\gamma)^3} + \frac{2C\epsilon_\Pi + 3\gamma^{K-1} V_{\max}}{(1-\gamma)^2} + \frac{V_{\max}\epsilon_\zeta}{1-\gamma}
$$

*for any policy $\widetilde{\pi} \in \Pi_C^{all}$ and only take action over $\mathcal{A}$.*

*Proof.* For any policy $\widetilde{\pi}$ that only take action over $\mathcal{A}$, Lemma 3 tells that $v_M^{\widetilde{\pi}} \leq v_{M'}^{\Xi(\widetilde{\pi})} + \frac{V_{\max}\epsilon_\zeta}{1-\gamma}$. Since $\pi_t = \Xi(\widehat{\pi}_t)$ and $\widehat{\pi}_t$ only takes action in $\mathcal{A}$, by Lemma 1 and Lemma 2 $v_M^{\widehat{\pi}_t} = v_{M'}^{\widehat{\pi}_t} \geq v_M^{\pi_t}$. Then $v_M^{\widetilde{\pi}} - v_M^{\widehat{\pi}_t} \leq v_{M'}^{\Xi(\widetilde{\pi})} - v_{M'}^{\pi_t} + \frac{V_{\max}\epsilon_\zeta}{1-\gamma}$ and Theorem 2 completes the proof. $\quad\square$

When there exist an optimal policy that is supported well by $\mu$. We can derive the following result about value gap between learned policy and optimal policy immediately from the main theorem about approximate policy iteration.

**Corollary 2.** *If there exists an $\pi^\star$ on $M$ such that $\Pr(\mu(s,a) \leq 2b|\pi^\star) \leq \epsilon$. then under the assumptions of Theorem 1, $\widehat{\pi}_t$ from Algorithm 1 satisfies that with probability at least $1 - 3\delta$,*

$$v_M^{\pi^\star} - v_M^{\pi_t} \leq \frac{4C}{(1-\gamma)^3} \left( \sqrt{\frac{419 V_{\max}^2 \ln \frac{|\mathcal{F}||\Pi|}{\delta}}{3n}} + 2\sqrt{\epsilon_{\mathcal{F}}} \right) + \frac{6CV_{\max}\epsilon_\mu}{(1-\gamma)^3}$$

$$+ \frac{2C\epsilon_\Pi + 3\gamma^{K-1}V_{\max}}{(1-\gamma)^2} + \frac{V_{\max}(\epsilon + C\epsilon_\mu)}{1-\gamma}$$

*Proof.* Given the condition of $\pi^\star$,

$$\Pr\left(\widehat{\mu}(s,a) \leq b \Big| \pi^\star\right) \leq \Pr\left(\mu(s,a) \leq 2b|\pi^\star\right) + \Pr\left(|\mu(s,a) - \widehat{\mu}(s,a)| \geq b|\pi^\star\right) \tag{93}$$

$$\leq \epsilon + \Pr\left(|\mu(s,a) - \widehat{\mu}(s,a)| \geq b|\pi^\star\right) \tag{94}$$

$$\leq \epsilon + \frac{\mathbb{E}_{\eta^{\pi^\star}}\left[|\mu(s,a) - \widehat{\mu}(s,a)|\right]}{b} \tag{95}$$

$$\leq \epsilon + \frac{U d_{\mathrm{TV}}(\mu(s,a), \widehat{\mu}(s,a))}{b} \tag{96}$$

$$\leq \epsilon + C\epsilon_\mu \tag{97}$$

Then $\pi^\star \in \Pi_C^{all}$ with $\epsilon_\zeta = \epsilon + C\epsilon_\mu$, and applying Theorem 1 finished the proof. $\qquad\square$

## C.5  Safe Policy Improvement Result

In many scenarios we aim to have a policy improvement that is guaranteed to be no worse than the data collection policy, which is called safe policy improvement. By abusing the notation a bit, let $\mu(a|s)$ be a policy that generate the data set. For our algorithm, the safe policy improvement will hold if $\mu \in \Pi_C^{all}$. To show $\mu \in \Pi_C^{all}$, we only need that $\Pr(\mu(s,a) \leq b|\mu) \leq \epsilon_\zeta$. When the state-action space is finite, there must exist an minimum value for all non-zero $\mu(s,a)$'s. Let $\mu_{\min} = \min_{s,a \, s.t. \mu(s,a)>0} \mu(s,a)$. Then we have that, if $b \leq \mu_{\min}$. $\Pr(\mu(s,a) \leq b|\mu) = 0$. Thus we have:

**Corollary 3.** *With finite state action space and $b \leq \mu_{\min}$, under the assumptions as Theorem 1, $\widehat{\pi}_t$ from Algorithm 1 satisfies that with probability at least $1 - 3\delta$,*

$$v_M^\mu - v_M^{\widehat{\pi}_t} \leq \frac{52 V_{\max}\sqrt{|\mathcal{S}||\mathcal{A}|}(\sqrt{\ln(2|\mathcal{S}||\mathcal{A}|/\delta)} + \sqrt{\ln(1 + nV_{\max})}) + 8}{\sqrt{n}b(1-\gamma)^3}$$

$$+ \frac{12 V_{\max}|\mathcal{S}||\mathcal{A}|\ln(2|\mathcal{S}||\mathcal{A}|/\delta)}{nb(1-\gamma)^3} + \frac{3\gamma^{K-1}V_{\max}}{(1-\gamma)^2}$$

*Proof.* In finite state action space, the number of all deterministic policies is less than $|\mathcal{A}|^{|\mathcal{S}|}$. Thus we have a policy class with $\epsilon_\Pi = 0$ and $|\Pi| \leq |\mathcal{A}|^{|\mathcal{S}|}$. Since the $Q$ value is bounded in $[0, V_{\max}]$, we can construct a $\epsilon$ covering set $\mathcal{F}$ of all value functions in $[0, V_{\max}]^{|\mathcal{S}||\mathcal{A}|}$ with $(\frac{V_{\max}}{\epsilon} + 1)^{|\mathcal{S}||\mathcal{A}|}$ functions. Then $\epsilon_{\mathcal{F}} \leq \max_g \min_{f \in \mathcal{F}} \|f - g\|_{\mu,2} \leq \max_g \min_{f \in \mathcal{F}} \|f - g\|_\infty \leq \epsilon$.

We can also bound $\epsilon_\mu$ in finite state action space. For any fixed $s, a$, by Berstein's inequality we have that with probability of $1 - \frac{\delta}{|\mathcal{S}||\mathcal{A}|}$:

$$|\widehat{\mu}(s,a) - \mu(s,a)| = \left| \frac{1}{n}\sum_{i=1}^n \mathbb{1}(s^{(i)} = s, a^{(i)} = a) - \mathbb{E}[\mathbb{1}(s^{(i)} = s, a^{(i)} = a)] \right| \tag{98}$$

$$\leq \sqrt{\frac{2\mathbb{V}[\mathbb{1}(s^{(i)} = s, a^{(i)} = a)]\ln(2|\mathcal{S}||\mathcal{A}|/\delta)}{n}} + \frac{4\ln(2|\mathcal{S}||\mathcal{A}|/\delta)}{n} \tag{99}$$

$$= \sqrt{\frac{2\mu(s,a)(1 - \mu(s,a))\ln(2|\mathcal{S}||\mathcal{A}|/\delta)}{n}} + \frac{4\ln(2|\mathcal{S}||\mathcal{A}|/\delta)}{n} \tag{100}$$

By taking summation of $|\widehat{\mu}(s,a) - \mu(s,a)|$ and union bound over all $(s,a)$, we can bound the total variation bounds between $\widehat{\mu}$ and $\mu$, with probability at least $1-\delta$,

$$\|\widehat{\mu} - \mu\|_{TV} = \frac{1}{2}\sum_{s,a} |\widehat{\mu}(s,a) - \mu(s,a)| \tag{101}$$

$$\leq \frac{1}{2}\sum_{s,a}\left(\sqrt{\frac{2\mu(s,a)(1-\mu(s,a))\ln(2|\mathcal{S}||\mathcal{A}|/\delta)}{n}} + \frac{4\ln(2|\mathcal{S}||\mathcal{A}|/\delta)}{n}\right) \tag{102}$$

$$= \frac{2|\mathcal{S}||\mathcal{A}|\ln(2|\mathcal{S}||\mathcal{A}|/\delta)}{n} + \frac{1}{2}\sum_{s,a}\sqrt{\frac{2\mu(s,a)(1-\mu(s,a))\ln(2|\mathcal{S}||\mathcal{A}|/\delta)}{n}} \tag{103}$$

$$\leq \frac{2|\mathcal{S}||\mathcal{A}|\ln(2|\mathcal{S}||\mathcal{A}|/\delta)}{n} + \frac{1}{2}\sqrt{\sum_{s,a}\frac{2\mu(s,a)\ln(2|\mathcal{S}||\mathcal{A}|/\delta)}{n}\sum_{s,a}(1-\mu(s,a))}$$

(Cauchy-Schwartz's inequality)

$$= \frac{2|\mathcal{S}||\mathcal{A}|\ln(2|\mathcal{S}||\mathcal{A}|/\delta)}{n} + \frac{1}{2}\sqrt{\frac{2\ln(2|\mathcal{S}||\mathcal{A}|/\delta)}{n}(|\mathcal{S}||\mathcal{A}|-1)} \tag{104}$$

$$\leq \frac{2|\mathcal{S}||\mathcal{A}|\ln(2|\mathcal{S}||\mathcal{A}|/\delta)}{n} + \sqrt{\frac{|\mathcal{S}||\mathcal{A}|\ln(2|\mathcal{S}||\mathcal{A}|/\delta)}{2n}} \tag{105}$$

Now in a finite state action space we can construct the policy and $Q$ function sets with $|\mathcal{F}| \leq (\frac{V_{\max}}{\epsilon}+1)^{|\mathcal{S}||\mathcal{A}|}$, $|\Pi| \leq |A|^{|S|}$, $\epsilon_\Pi = 0$, $\epsilon_\mathcal{F} \leq \epsilon$, and bounded $\epsilon_\mu$. By plugging these terms into the result of Theorem 1, we have the following bound:

$$v_M^\mu - v_M^{\hat{\pi}_t} \leq \frac{4C}{(1-\gamma)^3}\left(\sqrt{\frac{419V_{\max}^2(|\mathcal{S}|\ln|\mathcal{A}| + |\mathcal{S}||\mathcal{A}|\ln(1+V_{\max}/\epsilon) + \ln(1/\delta))}{3n}} + 2\sqrt{\epsilon}\right)$$
$$+ \frac{6CV_{\max}}{(1-\gamma)^3}\left(\sqrt{\frac{|\mathcal{S}||\mathcal{A}|\ln(2|\mathcal{S}||\mathcal{A}|/\delta)}{2n}} + \frac{2|\mathcal{S}||\mathcal{A}|\ln(2|\mathcal{S}||\mathcal{A}|/\delta)}{n}\right) + \frac{3\gamma^{K-1}V_{\max}}{(1-\gamma)^2}, \tag{106}$$

for any chosen $\epsilon > 0$. So we can set that $\epsilon = 1/n$ to upper bound the the infimum of this upper bound.

$$v_M^\mu - v_M^{\hat{\pi}_t} \leq \frac{4C}{(1-\gamma)^3}\left(\sqrt{\frac{419V_{\max}^2(|\mathcal{S}|\ln|\mathcal{A}| + |\mathcal{S}||\mathcal{A}|\ln(1+nV_{\max}) + \ln(1/\delta))}{3n}} + 2\sqrt{\frac{1}{n}}\right)$$
$$+ \frac{6CV_{\max}}{(1-\gamma)^3}\left(\sqrt{\frac{|\mathcal{S}||\mathcal{A}|\ln(2|\mathcal{S}||\mathcal{A}|/\delta)}{2n}} + \frac{2|\mathcal{S}||\mathcal{A}|\ln(2|\mathcal{S}||\mathcal{A}|/\delta)}{n}\right) + \frac{3\gamma^{K-1}V_{\max}}{(1-\gamma)^2} \tag{107}$$

Notice that in discrete space we have that $U \leq 1$. By replacing $C$ with $1/b$ and simplify some terms, we have that:

$$v_M^\mu - v_M^{\hat{\pi}_t} \leq \sqrt{\frac{6704V_{\max}^2|\mathcal{S}|(\ln(|\mathcal{A}|/\delta) + |\mathcal{A}|\ln(1+nV_{\max}))}{3nb^2(1-\gamma)^6}} + \frac{8}{b\sqrt{n}(1-\gamma)^3}$$
$$+ \sqrt{\frac{18V_{\max}^2|\mathcal{S}||\mathcal{A}|\ln(2|\mathcal{S}||\mathcal{A}|/\delta)}{nb^2(1-\gamma)^6}} + \frac{12V_{\max}|\mathcal{S}||\mathcal{A}|\ln(2|\mathcal{S}||\mathcal{A}|/\delta)}{nb(1-\gamma)^3} + \frac{3\gamma^{K-1}V_{\max}}{(1-\gamma)^2}$$
$$\leq \frac{52V_{\max}\sqrt{|\mathcal{S}||\mathcal{A}|}(\sqrt{\ln(2|\mathcal{S}||\mathcal{A}|/\delta)} + \sqrt{\ln(1+nV_{\max})}) + 8}{\sqrt{n}b(1-\gamma)^3}$$
$$+ \frac{12V_{\max}|\mathcal{S}||\mathcal{A}|\ln(2|\mathcal{S}||\mathcal{A}|/\delta)}{nb(1-\gamma)^3} + \frac{3\gamma^{K-1}V_{\max}}{(1-\gamma)^2}$$

$\square$

## D  Proofs for $Q$ Iteration Guarantees

In this section, we are going to prove the our main result for the $Q$ iteration algorithm, Algorithm 2. First we introduce a similar completeness assumption about the Bellman optimality operator:

**Assumption 5** (Completeness under $\mathcal{T}_\zeta$). $\max_{f \in \mathcal{F}} \min_{g \in \mathcal{F}} \|g - \mathcal{T}_\zeta f\|_{2,\mu}^2 \leq \epsilon_{\mathcal{F}}$

We will first state our main theorem here and then give a proof sketch before we start the proof formally.

**Theorem 4.** *Given a MDP $M = <\mathcal{S}, \mathcal{A}, R, P, \gamma, p>$, a dataset $D = \{(s, a, r, s')\}$ with $n$ samples that is draw i.i.d. from $\mu \times R \times P$, and a finite Q-function classes $\mathcal{F}$ satisfying Assumption 5, $\widehat{\pi}_t$ from Algorithm 2 satisfies that with probability at least $1 - \delta$, $v^{\widetilde{\pi}} - v^{\widehat{\pi}_t} \leq$*

$$\frac{2C}{(1-\gamma)^2}\left(\sqrt{\frac{208V_{\max}^2 \ln\frac{|\mathcal{F}|}{\delta}}{3n}} + 2\sqrt{\epsilon_{\mathcal{F}}} + V_{\max}\epsilon_\mu + \left\|Q^{\widetilde{\pi}} - \mathcal{T}_\zeta Q^{\widetilde{\pi}}\right\|_{2,\mu}\right) + \frac{(2\gamma^t + \epsilon_\zeta)V_{\max}}{1-\gamma}$$

*for any policy $\widetilde{\pi} \in \Pi_C^{all}$.*

We will first give a proof sketch before we start the proof formally. The proof follows a similar structural as the policy iteration case. To prove Theorem 4 we first prove a similar version of Theorem 4 but the comparator polices are in strong $\zeta$-constrained policy set (formally stated as Theorem 5 later). Then we show an upper bound of $v_{M'}^\pi - v_{M'}^{\pi_t}$, where $\pi \in \Pi_{SC}^{all}$ and $\pi_t$ is the output of algorithm (Theorem 5, will be formally stated later). Then we are going to show that for any policy $\pi$ in the $\zeta$-constrained policy set , after a projection $\Xi$ it is in the strong $\zeta$-constrained policy set and $v_M^\pi \leq v_{M'}^{\Xi(\pi)} + V_{\max}\epsilon_\zeta/(1-\gamma)$. Then we can provide the upper bound for $v_M^\pi - v_M^{\pi_t}$ for any $\pi$ in $\zeta$-constrained policy set (Theorem 4).

The proof sketch of Theorem 5 goes as follow. One key step to prove this error bound is to convert the performance difference between any policy $\widetilde{\pi} \in \Pi_{SC}^{all}$ and $\pi_t$ to a value function gap that is filtered by $\zeta$:

$$v^{\widetilde{\pi}} - v^{\pi_t} \leq \|\zeta\left(Q^{\widetilde{\pi}} - f_t\right)\|_{1,\nu_1}/(1-\gamma),$$

where $\nu_1$ is some admissible distribution over $\mathcal{S} \times \mathcal{A}$. The filter $\zeta$ allows the change of measure from $\nu_1$ to $\mu$ without constraining the density ratio between an arbitrary distribution $\nu$ and $\mu$. Instead for any $s, a$ where $\zeta$ is one, by definition $\mu$ is lower bounded and the density ratio is bounded by $C$ (details in Lemma 13).

The rest of the proof has a similar structure with the standard FQI analysis. In Lemma 15, we bound the norm $\|\zeta(Q^{\widetilde{\pi}} - f_t)\|_{2,\nu_1}$ by $C\|(f_t - \mathcal{T}_\zeta f_t)\|_{2,\mu}/(1-\gamma)$ and one additional sub-optimality error $\|Q^{\widetilde{\pi}} - \mathcal{T}_\zeta Q^{\widetilde{\pi}}\|_{2,\mu}$. The additional sub-optimality error term comes from the fact that $\widetilde{\pi}$ may not be an optimal policy since the optimal policy may not be a $\zeta$-constrained policy. The last step to finish the proof is to bound the expected Bellman residual by concentration inequality. Lemma 16 shows how to bound that following a similar approach as [1]. Then the main theorem is proved by combine all those steps. After that we prove when we can bound the value gap with resepct to optimal value in Corollary 4.

Now we start the proof. We are going to condition on the high probability bounds in Assumption 2 holds when we proof the lemmas.

**Lemma 13.** *For $\pi_t = \Xi(\widehat{\pi}_t)$ in Algorithm 2, for any policy $\widetilde{\pi} \in \Pi_{SC}^{all}$ we have*

$$v^{\widetilde{\pi}} - v^{\pi_t} \leq \sum_{h=0}^{\infty} \gamma^h \left(\left\|\zeta\left(Q^{\widetilde{\pi}} - f_t\right)\right\|_{1,\eta_h^{\pi_t} \times \widetilde{\pi}} + \left\|\zeta\left(Q^{\widetilde{\pi}} - f_t\right)\right\|_{1,\eta_h^\pi \times \pi_t}\right).$$

*Proof.* Given a deterministic greedy policy $\widehat{\pi}_t$, $\pi_t = \Xi(\widehat{\pi}_t)$ is also a deterministic policy and $\pi_t(s)$ equals $\widehat{\pi}_t(s)$ unless $\zeta(s, \widehat{\pi}_t(s)) = 0$, where $\pi_t(s) = a_{abs}$. Notice $\widehat{\pi}_t(s)$ is the maximizer of $\zeta(s, \cdot)f_t(s, \cdot)$. If $\zeta(s, \widehat{\pi}_t(s)) = 0$ then $\zeta(s, a)f_t(s, a) = 0$ for all $a$. We have that $\pi_t(s)$ is also the

maximizer of $\zeta(s,\cdot)f_t(s,\cdot)$.

$$v^{\widetilde{\pi}} - v^{\pi_t} = \sum_{h=0}^{\infty} \gamma^h \mathbb{E}_{s\sim\eta_h^{\pi_t}}[Q^{\widetilde{\pi}}(s,\widetilde{\pi}) - Q^{\widetilde{\pi}}(s,\pi_t)] \qquad\qquad\text{([3, Lemma 6.1])}$$

$$\leq \sum_{h=0}^{\infty} \gamma^h \mathbb{E}_{s\sim\eta_h^{\pi_t}}\left[\zeta(s,\widetilde{\pi})Q^{\widetilde{\pi}}(s,\widetilde{\pi}) - \zeta(s,\pi_t)Q^{\widetilde{\pi}}(s,\pi_t)\right] \qquad\qquad (108)$$

$$\leq \sum_{h=0}^{\infty} \gamma^h \mathbb{E}_{s\sim\eta_h^{\pi_t}}[\zeta(s,\widetilde{\pi})Q^{\widetilde{\pi}}(s,\widetilde{\pi}) - \zeta(s,\widetilde{\pi})f_t(s,\widetilde{\pi}) + \zeta(s,\pi_t)f_t(s,\pi_t) - \zeta(s,\pi_t)Q^{\widetilde{\pi}}(s,\pi_t)]$$

$$(109)$$

$$\leq \sum_{h=0}^{\infty} \gamma^h \left( \left\| \zeta\left(Q^{\widetilde{\pi}} - f_t\right) \right\|_{1,\eta_h^{\pi_t}\times\widetilde{\pi}} + \left\| \zeta\left(Q^{\widetilde{\pi}} - f_t\right) \right\|_{1,\eta_h^{\pi_t}\times\pi_t} \right) \qquad (110)$$

Equation (108) follows from the fact that for all $s,a$ such that $\widetilde{\pi}(a|s) > 0$, either $\zeta(s,a) = 1$, or $a = a_{\text{abs}}$. $a = a_{\text{abs}}$ indicates that $Q^{\widetilde{\pi}}(s,a) = 0$. So for all $s,a$ such that $\widetilde{\pi}(a|s) > 0$, $Q^{\widetilde{\pi}}(s,a) = \zeta(s,a)Q^{\widetilde{\pi}}(s,a)$. The second part follows from that for any $s,a$, $Q^{\widetilde{\pi}}(s,a) \geq \zeta(s,a)Q^{\widetilde{\pi}}(s,a)$. Equation (109) follows from the fact that $\pi_t(s)$ is the maximizer of $\zeta(s,\cdot)f_t(s,\cdot)$. $\qquad\square$

**Lemma 14.** *For any two function $f_1, f_2 : \mathcal{S}' \times \mathcal{A}' \to \mathbb{R}^+$, define $\pi_{f_1,f_2}(s) = \arg\max_{a\in\mathcal{A}} |f_1(s,a) - f_2(s,a)|$. Then we have $\forall\nu : \mathcal{S}' \to \Delta(\mathcal{A}')$,*

$$\left\| \max_{a\in\mathcal{A}} f_1 - \max_{a\in\mathcal{A}} f_2 \right\|_{1,P(\nu)} \leq \|f_1 - f_2\|_{1,P(\nu)\times\pi_{f_1,f_2}}.$$

*Proof.*

$$\left\| \max_{a\in\mathcal{A}} f_1 - \max_{a\in\mathcal{A}} f_2 \right\|_{1,P(\nu)} = \mathbb{E}_{s\sim P(\nu)}\left| \max_{a\in\mathcal{A}} f_1(s,a) - \max_{a\in\mathcal{A}} f_2(s,a) \right|$$

$$\leq \mathbb{E}_{s\sim P(\nu)} \max_{a\in\mathcal{A}} |f_1(s,a) - f_2(s,a)|$$

$$= \mathbb{E}_{s\sim P(\nu), a\sim\pi_{f_1,f_2}} |f_1(s,a) - f_2(s,a)|$$

$$= \|f_1 - f_2\|_{1,P(\nu)\times\pi_{f_1,f_2}}^2.$$

$\qquad\square$

**Lemma 15.** *For the data distribution $\mu$ and any admissible distribution $\nu$ over $\mathcal{S}' \times \mathcal{A}'$, $f, f' : \mathcal{S} \times \mathcal{A} \to \mathbb{R}^+$ and any $\widetilde{\pi} \in \Pi_{SC}^{all}$, we have*

$$\left\| \zeta\left(f - Q^{\widetilde{\pi}}\right) \right\|_{1,\nu} \leq C\left( \|f - \mathcal{T}_\zeta f'\|_{2,\mu} + \left\| \mathcal{T}_\zeta Q^{\widetilde{\pi}} - Q^{\widetilde{\pi}} \right\|_{2,\mu} + V_{\max}\epsilon_\mu \right)$$

$$+ \gamma \left\| \zeta\left(f' - Q^{\widetilde{\pi}}\right) \right\|_{2,P(\nu)\times\pi_{\zeta f',\zeta Q^{\widetilde{\pi}}}}.$$

*Proof.*

$$\left\| \zeta \left( f - Q^{\widetilde{\pi}} \right) \right\|_{1,\nu} \tag{111}$$

$$= \left\| \zeta \left( f - \mathcal{T}_\zeta f' + \mathcal{T}_\zeta f' - \mathcal{T}_\zeta Q^{\widetilde{\pi}} + \mathcal{T}_\zeta Q^{\widetilde{\pi}} - Q^{\widetilde{\pi}} \right) \right\|_{1,\nu} \tag{112}$$

$$\leq \left\| \zeta \left( f - \mathcal{T}_\zeta f' \right) \right\|_{1,\nu} + \left\| \zeta \left( \mathcal{T}_\zeta f' - \mathcal{T}_\zeta Q^{\widetilde{\pi}} \right) \right\|_{1,\nu} + \left\| \zeta \left( \mathcal{T}_\zeta Q^{\widetilde{\pi}} - Q^{\widetilde{\pi}} \right) \right\|_{1,\nu} \tag{113}$$

$$\leq C \left\| f - \mathcal{T}_\zeta f' \right\|_{1,\widehat{\mu}} + \gamma \left\| \max_{a \in \mathcal{A}} \zeta f' - \max_{a \in \mathcal{A}} \zeta Q^{\widetilde{\pi}} \right\|_{1,P(\nu)} + C \left\| \mathcal{T}_\zeta Q^{\widetilde{\pi}} - Q^{\widetilde{\pi}} \right\|_{1,\widehat{\mu}} \tag{114}$$

$$\leq 2C V_{\max} \epsilon_\mu + C \left\| f - \mathcal{T}_\zeta f' \right\|_{1,\mu} + \gamma \left\| \max_{a \in \mathcal{A}} \zeta f' - \max_{a \in \mathcal{A}} \zeta Q^{\widetilde{\pi}} \right\|_{1,P(\nu)} + C \left\| \mathcal{T}_\zeta Q^{\widetilde{\pi}} - Q^{\widetilde{\pi}} \right\|_{1,\mu} \tag{115}$$

$$\leq C \left( \left\| f - \mathcal{T}_\zeta f' \right\|_{2,\mu} + \left\| \mathcal{T}_\zeta Q^{\widetilde{\pi}} - Q^{\widetilde{\pi}} \right\|_{1,\mu} + 2 V_{\max} \epsilon_\mu \right) + \gamma \left\| \zeta \left( f' - Q^{\widetilde{\pi}} \right) \right\|_{1,P(\nu) \times \pi_{\zeta f', \zeta Q^{\widetilde{\pi}}}} \tag{116}$$

The change of norms from $\| \cdot \|_\nu$ to $\| \cdot \|_\mu$ follows from that $\zeta(s,a) \neq 0$ iff $\widehat{\mu}(s,a) \geq b$ and thus $\nu(s,a) \leq \widehat{\mu}(s,a) U / b = C \widehat{\mu}(s,a)$. The last step follows from Lemma 14. $\left\| \zeta \left( \mathcal{T}_\zeta f' - \mathcal{T}_\zeta Q^{\widetilde{\pi}} \right) \right\|_{1,\nu} \leq \gamma \left\| \max_{a \in \mathcal{A}} \zeta f' - \max_{a \in \mathcal{A}} \zeta Q^{\widetilde{\pi}} \right\|_{1,P(\nu)}$ follows from:

$$\left\| \zeta \left( \mathcal{T}_\zeta f' - \mathcal{T}_\zeta Q^{\widetilde{\pi}} \right) \right\|_{1,\nu} = \mathbb{E}_{(s,a) \sim \nu} \left[ \zeta(s,a) \left| \mathcal{T}_\zeta f'(s,a) - \mathcal{T}_\zeta Q^{\widetilde{\pi}}(s,a) \right| \right] \tag{117}$$

$$\leq \mathbb{E}_{(s,a) \sim \nu} \left[ \left| \mathcal{T}_\zeta f'(s,a) - \mathcal{T}_\zeta Q^{\widetilde{\pi}}(s,a) \right| \right] \tag{118}$$

$$= \mathbb{E}_{(s,a) \sim \nu} \left[ \left| \gamma \mathbb{E}_{s' \sim P(s,a)} \max_{a' \in \mathcal{A}} \zeta(s',a') f'(s',a') - \max_{a' \in \mathcal{A}} \zeta(s',a') Q^{\widetilde{\pi}}(s',a') \right| \right] \tag{119}$$

$$\leq \gamma \, \mathbb{E}_{(s,a) \sim \nu, s' \sim P(s,a)} \left[ \left| \max_{a' \in \mathcal{A}} \zeta(s',a') f'(s',a') - \max_{a' \in \mathcal{A}} \zeta(s',a') Q^{\widetilde{\pi}}(s',a') \right| \right] \tag*{(Jensen)}$$

$$= \gamma \, \mathbb{E}_{s' \sim P(\nu)} \left[ \left| \max_{a' \in \mathcal{A}} \zeta(s',a') f'(s',a') - \max_{a' \in \mathcal{A}} \zeta(s',a') Q^{\widetilde{\pi}}(s',a') \right| \right] \tag{120}$$

$$= \gamma \left\| \max_{a \in \mathcal{A}} \zeta f' - \max_{a \in \mathcal{A}} \zeta Q^{\widetilde{\pi}} \right\|_{1,P(\nu)} \tag{121}$$

$\square$

Now we are going to use Berstein's inequality to bound $\| f_{t+1} - \mathcal{T}_\zeta f_t \|_{2,\mu}$, which mostly follows from [1]'s proof for the vanilla value iteration.

**Lemma 16.** *With Assumption 5 holds, let $g_f^\star = \arg\min_{g \in \mathcal{F}} \| g - \mathcal{T}_\zeta f \|_{2,\mu}$, then $\| g_f^\star - \mathcal{T}_\zeta f \|_{2,\mu}^2 \leq \epsilon_{\mathcal{F}}$.*
*The dataset $D$ is generated i.i.d. from $M$ as follows: $(s,a) \sim \mu$, $r = R(s,a)$, $s' \sim P(s,a)$. Define $\mathcal{L}_\mu(f; f') = \mathbb{E}[\mathcal{L}_D(f; f')]$. We have that $\forall f \in \mathcal{F}$, with probability at least $1 - \delta$,*

$$\mathcal{L}_\mu(\mathcal{T}_{\zeta,D} f; f) - \mathcal{L}_\mu(g_f^\star; f) \leq \frac{208 V_{\max}^2 \ln \frac{|\mathcal{F}|}{\delta}}{3n} + \epsilon_{\mathcal{F}}$$

*where $\mathcal{T}_{\zeta,D} f = \arg\min_{g \in \mathcal{F}} \mathcal{L}_D(g, f)$.*

*Proof.* This proof is similar with the proof of Lemma 7, and we adapt it to operator $\mathcal{T}_\zeta$. The only change is the definition of $V_f(\cdot)$ and $X(\cdot, \cdot, \cdot)$. The definition of $\mathcal{L}_D$ and $\mathcal{L}_\mu$ would not change between $M$ and $M'$, and the right hand side is also the same constant for $M$ and $M'$. So the result we prove here does not change from $M$ to $M'$.

For the simplicity of notations, let $V_f(s) = \max_{a \in \mathcal{A}} \zeta(s,a) f(s,a)$. Fix $f, g \in \mathcal{F}$, and define

$$X(g, f, g_f^\star) := \left( g(s,a) - r - \gamma V_f(s') \right)^2 - \left( g_f^\star(s,a) - r - \gamma V_f(s') \right)^2.$$

Plugging each $(s, a, r, s') \in D$ into $X(g, f, g_f^\star)$, we get i.i.d. variables $X_1(g, f, g_f^\star), X_2(g, f, g_f^\star), \ldots, X_n(g, f, g_f^\star)$. It is easy to see that

$$\frac{1}{n} \sum_{i=1}^{n} X_i(g, f, g_f^\star) = \mathcal{L}_D(g; f) - \mathcal{L}_D(g_f^\star; f).$$

By the definition of $\mathcal{L}_\mu$, it is also easy to see that

$$\mathcal{L}_\mu(g; f) = \|g - \mathcal{T}_\zeta f\|_{2,\mu}^2 + \mathbb{E}_{s,a \sim \mu} \left[ \mathbb{V}_{r,s'} \left( r + \gamma \max_{a' \in \mathcal{A}} \zeta(s', a') f(s', a') \right) \right]$$

Notice that the second part does not depends on $g$. Then

$$\mathcal{L}_\mu(g; f) - \mathcal{L}_\mu(\mathcal{T}_\zeta f; f) = \|g - \mathcal{T}_\zeta f\|_{2,\mu}^2$$

342 Then we bound the variance of $X$:

$$\mathbb{V}[X(g, f, g_f^\star)] \leq \mathbb{E}[X(g, f, g_f^\star)^2]$$
$$= \mathbb{E}_\mu \left[ \left( \left(g(s, a) - r - \gamma V_f(s')\right)^2 - \left(g_f^\star(s, a) - r - \gamma V_f(s')\right)^2 \right)^2 \right]$$
$$= \mathbb{E}_\mu \left[ \left(g(s, a) - g_f^\star(s, a)\right)^2 \left(g(s, a) + g_f^\star(s, a) - 2r - 2\gamma V_f(s')\right)^2 \right]$$
$$\leq 4 V_{\max}^2 \, \mathbb{E}_\mu \left[ \left(g(s, a) - g_f^\star(s, a)\right)^2 \right]$$
$$= 4 V_{\max}^2 \|g - g_f^\star\|_{2,\mu}^2 \tag{122}$$
$$\leq 8 V_{\max}^2 \left( \mathbb{E}[X(g, f, g_f^\star)] + 2\epsilon_\mathcal{F} \right). \tag{$*$}$$

343 Step (*) holds because

$$\|g - g_f^\star\|_{2,\mu}^2$$
$$\leq 2 \left( \|g - \mathcal{T}_\zeta f\|_{2,\mu}^2 + \|\mathcal{T}_\zeta f - g_f^\star\|_{2,\mu}^2 \right) \qquad ((a+b)^2 \leq 2a^2 + 2b^2)$$
$$\leq 2 \left( \|g - \mathcal{T}_\zeta f\|_{2,\mu}^2 - \|\mathcal{T}_\zeta f - g_f^\star\|_{2,\mu}^2 + 2\|\mathcal{T}_\zeta f - g_f^\star\|_{2,\mu}^2 \right)$$
$$= 2 \left[ \left( \mathcal{L}_\mu(g; f) - \mathcal{L}_\mu(\mathcal{T}_\zeta f; f) \right) - \left( \mathcal{L}_\mu(g_f^\star; f) - \mathcal{L}_\mu(\mathcal{T}_\zeta f; f) \right) + 2\|\mathcal{T}_\zeta f - g_f^\star\|_{2,\mu}^2 \right]$$
$$= 2 \left[ \left( \mathcal{L}_\mu(g; f) - \mathcal{L}_\mu(g_f^\star; f) \right) + 2\|\mathcal{T}_\zeta f - g_f^\star\|_{2,\mu}^2 \right]$$
$$= 2 \left( \mathbb{E}[X(g, f, g_f^\star)] + 2\|\mathcal{T}_\zeta f - g_f^\star\|_{2,\mu}^2 \right)$$
$$\leq 2 \left( \mathbb{E}\left[ X(g, f, g_f^\star) \right] + 2\epsilon_\mathcal{F} \right)$$

344 Next, we apply (one-sided) Bernstein's inequality and union bound over all $f \in \mathcal{F}$ and $g \in \mathcal{F}$. With
345 probability at least $1 - \delta$, we have

$$\mathbb{E}[X(g, f, g_f^\star)] - \frac{1}{n} \sum_{i=1}^{n} X_i(g, f, g_f^\star) \leq \sqrt{\frac{2 \mathbb{V}[X(g, f, g_f^\star)] \ln \frac{|\mathcal{F}|^2}{\delta}}{n}} + \frac{4 V_{\max}^2 \ln \frac{|\mathcal{F}|^2}{\delta}}{3n}$$

$$= \sqrt{\frac{32 V_{\max}^2 \left( \mathbb{E}[X(g, f, g_f^\star)] + 2\epsilon_\mathcal{F} \right) \ln \frac{|\mathcal{F}|}{\delta}}{n}} + \frac{8 V_{\max}^2 \ln \frac{|\mathcal{F}|}{\delta}}{3n}$$

Since $\mathcal{T}_{\zeta,D} f$ minimizes $\mathcal{L}_D(\cdot; f)$, it also minimizes $\frac{1}{n} \sum_{i=1}^{n} X_i(\cdot, f, g_f^\star)$. This is because the two objectives only differ by a constant $\mathcal{L}_D(g_f^\star; f)$. Hence,

$$\frac{1}{n} \sum_{i=1}^{n} X_i(\mathcal{T}_{\zeta,D} f, f, g_f^\star) \leq \frac{1}{n} \sum_{i=1}^{n} X_i(g_f^\star, f, g_f^\star) = 0.$$

346 Then,

$$\mathbb{E}[X(\mathcal{T}_{\zeta,D} f, f, g_f^\star)] \leq \sqrt{\frac{32 V_{\max}^2 \left( \mathbb{E}[X(\mathcal{T}_{\zeta,D} f, f, g_f^\star)] + 2\epsilon_\mathcal{F} \right) \ln \frac{|\mathcal{F}|}{\delta}}{n}} + \frac{8 V_{\max}^2 \ln \frac{|\mathcal{F}|}{\delta}}{3n}.$$

347 Solving for the quadratic formula,

$$\mathbb{E}[X(\mathcal{T}_{\zeta,D}f,f,g_f^\star)] \le \sqrt{48\left(\frac{8V_{\max}^2 \ln\frac{|\mathcal{F}|}{\delta}}{3n}\right)^2 + \frac{64V_{\max}^2 \ln\frac{|\mathcal{F}|}{\delta}}{n}\epsilon_{\mathcal{F}} + \frac{56V_{\max}^2 \ln\frac{|\mathcal{F}|}{\delta}}{3n}}$$

$$\le \frac{(56+32\sqrt{3})V_{\max}^2 \ln\frac{|\mathcal{F}|}{\delta}}{3n} + \sqrt{\frac{64V_{\max}^2 \ln\frac{|\mathcal{F}|}{\delta}}{n}\epsilon_{\mathcal{F}}}$$

$$(\sqrt{a+b} \le \sqrt{a} + \sqrt{b} \text{ and } \ln\tfrac{|\mathcal{F}|}{\delta} > 0)$$

$$\le \frac{112V_{\max}^2 \ln\frac{|\mathcal{F}|}{\delta}}{3n} + \sqrt{\frac{64V_{\max}^2 \ln\frac{|\mathcal{F}|}{\delta}}{n}\epsilon_{\mathcal{F}}}$$

$$\le \frac{112V_{\max}^2 \ln\frac{|\mathcal{F}|}{\delta}}{3n} + \frac{32V_{\max}^2 \ln\frac{|\mathcal{F}|}{\delta}}{n} + \epsilon_{\mathcal{F}}$$

$$\le \frac{208V_{\max}^2 \ln\frac{|\mathcal{F}|}{\delta}}{3n} + \epsilon_{\mathcal{F}}$$

348 Noticing that $\mathbb{E}[X(\mathcal{T}_{\zeta,D}f; f, g_f^\star)] = \mathcal{L}_\mu(\mathcal{T}_{\zeta,D}f; f) - \mathcal{L}_\mu(g_f^\star; f)$, we complete the proof. $\qquad\square$

349 Now we could prove the main theorem about fitted Q iteration.

350 **Theorem 5.** *Given a MDP $M =< \mathcal{S}, \mathcal{A}, R, P, \gamma, p >$, a dataset $D = \{(s,a,r,s')\}$ with $n$ samples*
351 *that is draw i.i.d. from $\mu \times R \times P$, and a finite Q-function classes $\mathcal{F}$ satisfying Assumption 5,*
352 *$\pi_t = \Xi(\widehat{\pi}_t)$ from Algorithm 2 satisfies that with probability at least $1 - 2\delta$, $v^{\widetilde{\pi}} - v^{\pi_t} \le$*

$$\frac{2C}{(1-\gamma)^2}\left(\sqrt{\frac{208V_{\max}^2 \ln\frac{|\mathcal{F}|}{\delta}}{3n}} + 2\sqrt{\epsilon_{\mathcal{F}}} + V_{\max}\epsilon_\mu + \left\|Q^{\widetilde{\pi}} - \mathcal{T}_\zeta Q^{\widetilde{\pi}}\right\|_{1,\mu}\right) + \frac{2\gamma^t V_{\max}}{1-\gamma}$$

353 *for any policy $\widetilde{\pi} \in \Pi_{SC}^{all}$.*

*Proof.* Firstly, we can let $f = f_t$ and $f' = f_{t-1}$ in Lemma 15. This gives us that

$$\left\|f_t - Q^{\widetilde{\pi}}\right\|_{1,\nu} \le C\left(\|f_t - \mathcal{T}_\zeta f_{t-1}\|_{2,\mu} + \left\|Q^{\widetilde{\pi}} - \mathcal{T}_\zeta Q^{\widetilde{\pi}}\right\|_{1,\mu} + 2V_{\max}\epsilon_\mu\right) + \gamma\|f_{t-1} - Q^{\widetilde{\pi}}\|_{1,P(\nu)\times\pi_{f_{k-1},Q^{\widetilde{\pi}}}}$$

354 Note that we can apply the same analysis on $P(\nu) \times \pi_{f_{k-1},Q^\star}$ and expand the inequality $t$ times. It
355 then suffices to upper bound $\|f_t - \mathcal{T}_\zeta f_{t-1}\|_{2,\mu}$.

$$\|f_t - \mathcal{T}_\zeta f_{t-1}\|_{2,\mu}^2$$
$$= \mathcal{L}_\mu(f_t; f_{t-1}) - \mathcal{L}_\mu(\mathcal{T}_\zeta f_{t-1}; f_{t-1}) \qquad\qquad (\text{Definition of } \mathcal{L}_\mu)$$
$$= [\mathcal{L}_\mu(f_t; f_{t-1}) - \mathcal{L}_\mu(g_{f_{t-1}}^\star; f_{t-1})] + [\mathcal{L}_\mu(g_{f_{t-1}}^\star; f_{t-1}) - \mathcal{L}_\mu(\mathcal{T}_\zeta f_{t-1}; f_{t-1})]$$
$$\le \epsilon_4 + \|g_{f_{t-1}}^\star - \mathcal{T}_\zeta f_{t-1}\|_{2,\mu}^2 \qquad\qquad (\text{Lemma 16 and definition of } \mathcal{L}_\mu)$$
$$\le \epsilon_4 + \epsilon_{\mathcal{F}}. \qquad\qquad (\text{Definition of } g_{Q_{k-1}}^\star \text{ and Assumption 5})$$

The inequality holds with probability at least $1 - \delta$ and $\epsilon_4 = \frac{208V_{\max}^2 \ln\frac{|\mathcal{F}|}{\delta}}{3n} + \epsilon_{\mathcal{F}}$. Noticing that $\epsilon_4$
and $\epsilon_{\mathcal{F}}$ do not depend on $t$, and the inequality holds simultaneously for different $t$, we have that

$$\|f_t - Q^{\widetilde{\pi}}\|_{1,\nu} \le \frac{1-\gamma^t}{1-\gamma}C\left(\sqrt{(\epsilon_4 + \epsilon_{\mathcal{F}})} + V_{\max}\epsilon_\mu + \left\|Q^{\widetilde{\pi}} - \mathcal{T}_\zeta Q^{\widetilde{\pi}}\right\|_{1,\mu}\right) + \gamma^t V_{\max}.$$

Applying this to Lemma 13, we have that

$$
v^{\widetilde{\pi}} - v^{\pi_t}
$$

$$
\leq \frac{2}{1-\gamma}\left(\frac{1-\gamma^t}{1-\gamma}C\left(\sqrt{(\epsilon_4+\epsilon_{\mathcal{F}})}+V_{\max}\epsilon_\mu+\left\|Q^{\widetilde{\pi}}-\mathcal{T}_\zeta Q^{\widetilde{\pi}}\right\|_{1,\mu}\right)+\gamma^t V_{\max}\right)
$$

$$
\leq \frac{2C}{(1-\gamma)^2}\left(\sqrt{\epsilon_4+\epsilon_{\mathcal{F}}}+V_{\max}\epsilon_\mu+\left\|Q^{\widetilde{\pi}}-\mathcal{T}_\zeta Q^{\widetilde{\pi}}\right\|_{1,\mu}\right)+\frac{2\gamma^t V_{\max}}{1-\gamma}
$$

$$
\leq \frac{2C}{(1-\gamma)^2}\left(\sqrt{\frac{208V_{\max}^2\ln\frac{|\mathcal{F}|}{\delta}}{3n}}+2\sqrt{\epsilon_{\mathcal{F}}}+V_{\max}\epsilon_\mu+\left\|Q^{\widetilde{\pi}}-\mathcal{T}_\zeta Q^{\widetilde{\pi}}\right\|_{1,\mu}\right)+\frac{2\gamma^t V_{\max}}{1-\gamma}.
$$

$\square$

Now we are going to use the fact that there is an no-value-loss projection from the $\zeta$-constrained policy set to the strong $\zeta$-constrained policy set to prove an error bound w.r.t any $\widetilde{\pi}\in\Pi_C^{all}$.

**Theorem 2.** *Given a MDP $M =< \mathcal{S},\mathcal{A},R,P,\gamma,p >$, a dataset $D = \{(s,a,r,s')\}$ with $n$ samples that is draw i.i.d. from $\mu\times R\times P$, and a finite Q-function classes $\mathcal{F}$ satisfying Assumption 5, $\widehat{\pi}_t$ from Algorithm 2 satisfies that with probability at least $1-2\delta$, $v^{\widetilde{\pi}}-v^{\widehat{\pi}_t}\leq$*

$$
\frac{2C}{(1-\gamma)^2}\left(\sqrt{\frac{208V_{\max}^2\ln\frac{|\mathcal{F}|}{\delta}}{3n}}+2\sqrt{\epsilon_{\mathcal{F}}}+V_{\max}\epsilon_\mu+\left\|Q^{\widetilde{\pi}}-\mathcal{T}_\zeta Q^{\widetilde{\pi}}\right\|_{2,\mu}\right)+\frac{(2\gamma^t+\epsilon_\zeta)V_{\max}}{1-\gamma}
$$

*for any policy $\widetilde{\pi}\in\Pi_C^{all}$.*

*Proof.* The difference between this theorem and Theorem 5 is that $\widetilde{\pi}$ is in $\Pi_C^{all}$ which is significantly larger than $\Pi_{SC}^{all}$.

This prove mimics the proof of Theorem 1. For any policy $\widetilde{\pi}\in\Pi_C^{all}$, Lemma 3 tells that $v_M^{\widetilde{\pi}}\leq v_{M'}^{\Xi(\widetilde{\pi})}+\frac{V_{\max}\epsilon_\zeta}{1-\gamma}$. Since $\pi_t=\Xi(\widehat{\pi}_t)$, $v_M^{\widehat{\pi}_t}=v_{M'}^{\widehat{\pi}_t}\geq v_M^{\pi_t}$. Then $v_M^{\widetilde{\pi}}-v_M^{\widehat{\pi}_t}\leq v_{M'}^{\Xi(\widetilde{\pi})}-v_{M'}^{\pi_t}+\frac{V_{\max}\epsilon_\zeta}{1-\gamma}$ and Theorem 5 completes the proof. $\square$

**Remark:** The first term in the theorem comes from that the best policy in the $\zeta$-constrained policy set is not optimal. Note that the $\zeta$-constrained policy set does not requires any realizability to do with our function approximation but merely about the density ratio of a policy. When there is an optimal policy of $M$ such in $\Pi_C^{all}$, we have the same type of bound as standard approximate value iteration analysis.

**Corollary 4.** *If there exists an $\pi^\star$ on $M$ such that $\Pr(\mu(s,a)\leq 2b|\pi^\star)\leq\epsilon$. then under the condition as Theorem 4, $\widehat{\pi}_t$ from Algorithm 2 satisfies that with probability at least $1-2\delta$, $v_M^{\pi^\star}-v_M^{\pi_t}\leq$*

$$
\frac{2C}{(1-\gamma)^2}\left(\sqrt{\frac{208V_{\max}^2\ln\frac{|\mathcal{F}|}{\delta}}{3n}}+2\sqrt{\epsilon_{\mathcal{F}}}+V_{\max}\epsilon_\mu+\left\|Q^{\pi^\star}-\mathcal{T}_\zeta Q^{\pi^\star}\right\|_{2,\mu}\right)+\frac{V_{\max}(2\gamma^t+\epsilon+CU\epsilon_\mu)}{1-\gamma}
$$

*Proof.* The proof of $\pi^\star\in\Pi_C^{all}$ is same as the proof in Corollary 1. Then proof is finished by applying Theorem 4. $\square$

# E  Details of CartPole Experiment

## E.1  Full results of Discretized CartPole-v0

In section 6.1, we compare AVI, BCQL[2], SPIBB[4], Behavior cloning and our algorithm PQI, in CartPole-v0 with discretized state space. The data is generated by a $\epsilon$-greedy policy ($\epsilon$ from 0.1 to 0.9) and we report the resulting policies from different algorithm with the best hyper-parameter in each $\epsilon$. In this section we show the learning curve for each $\epsilon$ and each hyper-parameter value. We run the BCQ algorithm with the threshold of $\widehat{\mu}(a|s)$ in $\{0, 0.05, 0.1, 0.2\}$, and we run the SPIBB

Figure 1: CartPole-v0 with discretized state space. The learning curve of all algorithms with different hyper-parameters, data generated with different $\epsilon$-greedy behavior policy. The hyper-parameter of SPIBB [4] and PQI is the threshold of $\widehat{\mu}(s, a)$ and the hyper-parameter of BCQL [2] is the threshold of $\widehat{\mu}(a|s)$.

algorithm with the threshold of $\widehat{\mu}(s, a)$ in $\{0.01, 0.005, 0.001, 0.0005, 0.0001\}$ and PQI with the threshold of $\widehat{\mu}(s, a)$ in a smaller set $\{0.005, 0.001, 0.0005\}$. Figure 1 shows for most of the $\epsilon$ and threshold our algorithm tie with the best baseline (SPIBB), and the best threshold of our algorithm outperform all baseline algorithms in 8 out of 9 cases.

In Figure 1, we observe the trend that smaller $\epsilon$ will prefer a smaller $b$. This is verified by more results in the next section, and we discuss the reasons for this phenomenon there.

## E.2  Ablation study of threshold b

A key aspect of our algorithm is to filter the state space by a threshold on the estimated probability $\widehat{\mu}(s, a)$. This prevents the algorithm from updating using low-confidence state, action pairs when bootstrapping values. Then the choice of threshold $b$ is a key trade-off in our algorithm: if $b$ is too small it can not remove the low-confident state, action pairs effectively; if $b$ is too large it might remove too many state, action pairs and prevent learning from more data. In order to demonstrate the effect of $b$ and how should we choose b in different settings, we show the performance of PQI in a larger range of $b$ and several $\epsilon$ values.

In figure 2 we show the trend that smaller $b$ works better for larger $\epsilon$ and larger $b$ works better for smaller $\epsilon$ in general. This can be explained in the following way: with a larger $\epsilon$ the data distribution is more exploratory and hence the probabilities on individual state, action pairs are smaller. So a the same threshold that performs well with low exploration now censors a much larger part of the state, action space, necessitating a smaller threshold as $\epsilon$ is increased. In general, we find that having the largest threshold which still retains a significant fraction of the state, action space is a good heuristic for setting the $b$ parameter.

Figure 2: Performance of PQI with different values of threshold $b$

## F Details of D4RL Experiment

In this section we introduce some missing details about the PQL algorithm and the experimental details in D4RL tasks. Our code is available at https://github.com/yaoliucs/PQL.

PQL algorithm is implemented based on the architecture of Batch-Constrained deep $Q$-learning (BCQ) [2] algorithm. More specifically, we use the similar Clipped Double Q-Learning (CDQ) update rule for the $Q$ learning part, and employ a similar variational auto-encoder to fit the conditional action distribution in the batch. We use an additional variational auto-encoder to fit the marginalized state distribution of the batch. To implement an actual $Q$ learning algorithm instead of an actor-critic algorithm, we did not sample from the actor in the Bellman backup but sample a larger batch from the fitted conditional action distribution. Algorithm 4 shows the pseudo-code of PQL to provide more details. We highlight the difference with the BCQ algorithm in red.

---

**Algorithm 4** Pessimistic $Q$-learning (PQL)

---

**Input:** Batch $D$, ELBO threshold $b$, maximum perturbation $\Phi$, target update rate $\tau$, mini-batch size $N$, max number of iteration $T$. Number of actions $k$.
Initialize two Q network $Q_{\theta_1}$ and $Q_{\theta_2}$, policy (perturbation) model: $\xi_\phi$. ($\xi_\phi \in [-\Phi, \Phi]$), action VAE $G^a_{\omega_1}$ and state VAE $G^s_{\omega_2}$.
Pretrain $G^s_{\omega_2}$: $\omega_2 \leftarrow \arg\min_{\omega_2} ELBO(B; G^s_{\omega_2})$.
**for** $t = 1$ **to** $T$ **do**
  Sample a minibatch $B$ with $N$ samples from $D$.
  $\omega_1 \leftarrow \arg\min_{\omega_2} ELBO(B; G^a_{\omega_1})$.
  Sample $k$ actions $a'_i$ from $G^a_{\omega_1}(s')$ for each $s'$.
  Compute the target $y$ for each $(s, a, r, s')$ pair:

$$y = r + \gamma \mathbb{1}(ELBO(s'; G^s_{\omega_2}) \geq b) \left[ \max_{a'_i} \left( 0.75 * \min_{j=1,2} Q_{\theta'_j} + 0.25 * \max_{j=1,2} Q_{\theta'_j} \right) \right]$$

  $\theta \leftarrow \arg\min_\theta \sum (y - Q_\theta(s, a))^2$
  Sample $k$ actions $a_i$ from $G^a_{\omega_1}(s)$ for each $s$.
  $\phi \leftarrow \arg\max_\phi \sum \max_{a_i} Q_{\theta_1}(s, a_i + \xi_\phi(s, a_i))$
  Update target network: $\theta' = (1 - \tau)\theta' + \tau\theta$, $\phi' = (1 - \tau)\phi' + \tau\phi$
**end for**
**When evaluate the resulting policy:** select action $a = \arg\max_{a_i} Q_{\theta_1}(s, a_i + \xi_\phi(s, a_i))$ where $a_i$ are $k$ actions sampled from $G^a_{\omega_1}(s)$ given $s$.

---

In practice, the indicator function $\mathbb{1}(ELBO(s'; G^s_{\omega_2}) \geq b)$ is implemented by $\text{sigmoid}(100(ELBO(s'; G^s_{\omega_2}) - b))$ to provide a slightly more smooth target. The evidence lower bound (ELBO) in VAE is:

$$ELBO(s; G^s_{\omega_2}) = \sum (s - \tilde{s})^2 + D_{\text{KL}}(N(\mu, \sigma) || N(0, 1)) \tag{123}$$

where $\mu$ and $\sigma$ is sampled from the encoder of VAE with input $s$ and $\tilde{s}$ is sampled from the decoder with the hidden state generated from $N(\mu, \sigma)$. $ELBO(B; G^s_{\omega_2})$ is the averaged ELBO on the minibatch $B$. So does $G^a_{\omega_1}$. Note that this ELBO objective make the implicit assumption that the decoder's distribution is a Gaussian distribution with mean equals to the output of decoder network.

So when we generate the sample $a'$ for computing $y$, we add a Gaussian noise to recover a sample from the full posterior distribution.

For most of the hyper-parameters in Algorithm 4, we use the same value with the BCQ algorithm. We run all algorithms with $T = 5 \times 10^5$ gradient steps as other reported results in D4RL tasks, and the minibatch size $N = 100$ at each step. The number of sampled action when running the policy is $k = 100$. Target network update rate is $0.005$. The threshold $b$ of ELBO is selected as 2-percentile of the $ELBO(s)$ in the whole dataset after pretrain the VAE.