[Reviews · NeurIPS 2020]

Review 1

Summary and Contributions: This paper proposes an algorithm for batch reinforcement learning which performs Bellman backups masked by the density of state-action pairs in the dataset distribution. The authors provide an analysis of their approach and provide some convincing evidence of their approach on some toy MDPs, Cartpole and Hopper.

Strengths: The work is interesting since it provides a new algorithm that prevents erroneous backups based on a masking multiplier in a regular fitted Q backup. They also provide analysis of their method and show some empirical results.

Weaknesses: I also feel that the paper could have benefited from a discussion of these as compared to just outrightly saying that existing methods do not give us good results. In particular, the conditions under which existing methods work vs do not work should have been discussed more explicitly than what it is right now in the paper. Moreover, I think the experiments on cartpole and hopper are not indicative of their method's performance since these have determnisitc dynamics and the dataset was collected as trajectories (so s' is as frequent as s in the distribution \mu, see my point below) and hence their choice of masking reduces to action conditioned masking only. Some other questions that I have: - From the analysis perspective, the paper says that prior works such as Kumar et al. 2019 that use action conditional and concentrability do not get the same error rate. Is the main issue behind this limitation that the notion of concentrability used in Kumar et al. and other works is trajectory centric and not on the state-action marginal? The latter was shown to be better than trajectory-based concentrability in Chen and Jiang (2019). If this notion of concentrability is used, would that be sufficient to get rid of the concentrability assumptions in your work. - Why does the method help on Hopper, which has deterministic dynamics, so given (s, a), there is a unique s', and in this case, it simply reduces to action-conditional masking? Can it be evaluated on some other domains with non-deterministic dynamics to evaluate its empirical efficacy? Otherwise empirically it seems like the method doesn't seem to have much benefit. Why is BEAR missing from baselines? - It seems like when the batch of data is a set of trajectories, and the state space is continuous, then the density of s_t is the same as s_{t+1} which is 1/N, so in that case does the proposed algorithm which exploits the fact that s_{t+1} may be highly infrequent compared to s_{t} reduce simply to an action conditional? The experiments are done with this setup too it seems. - Building on the previous point, if the data comes from d^\mu: the state visitation distribution of a behavior policy \mu, then d^\mu(s) and d^\mu(s') for a transition (s, a, r, s') observed in the dataset shouldn't be very different, in that case, would the proposed method be not (much) better than action-conditional penalty methods that have mostly been studied in this domain? - How do you compare algorithms, theoretically, for different values of b, and the hyperparameters for other algorithms, such as \eps in BEAR? It seems like for some value of both of these quantities, the algorithms should perform safely and not have much error accumulation. So, how is the proposed method better theoretically than the best configuration of the prior methods? - How will the proposed method compare to residual gradient algorithms which have better guarantees?

Correctness: The claims seem correct.

Clarity: The paper is well written.

Relation to Prior Work: Yes, relation to prior work is discussed well. Though, I am not certain if some claims against prior works are well justified or not.

Reproducibility: Yes

Additional Feedback: My comments are in my answer to a previous question. Few other questions - Is it possible to concretely elaborate on Lines 224-227? Why are guarantees not recovered? What part of the analysis breaks in this case? - Is it possible to comment on the error accumulation vs optimality tradeoff of the proposed method as compared to prior methods, such as BEAR? Seems like a stronger restriction is being placed on the policy class, and so, there is certainly less scope for improvment. One can trade off this for some error accumulation, so how does the proposed method stand in comparison? ------------------ Post discussion: I will retain my score, and I would strongly request authors to improve on the points I mention below. These will greatly improve readability. Thanks for the clarifications. I would appreciate more discussion on how exactly the concentration coefficients are gone, and what term instead compensates for that. Lines 219-227 compare the bound in theorem 1 to other prior works, but I am not sure if they provide enough intuition for how exactly the bound in Theorem 1 works. For example, some questions that I have, which would be good to have include: if this method removes the concentrability coefficient by restriction to the dataset, can other methods that clamp the Q-value or Bellman error using R_{max}/1 - \gamma (so that Q-values and Bellman error are meaningful and not too large, which is what is dictated by concentrability bounds when faced with distribution shift) also avoid the concentrability term and get a comparable bound. These questions might be silly but discussing these might help improve the intuition part of the paper very much, in my opinion. Also, a discussion of why there's an additional factor of (1 - gamma) as compared to Theorem 11 in the Chen and Jiang 2019 paper: https://arxiv.org/abs/1905.00360) should be discussed. How exactly does this additional horizon factor affect concentrability? As in if I treat the effective concentrability of the proposed algorithm in Theorem 1 in your paper as C/ (1 - \gamma) then even this depends on the horizon in addition to U and b. Typical notions of concentrability that go as C = d^\pi/ d^\mu also depend on a factor of horizon in the denominator, plus a per step error like term that I think can be shown to be a hyperparameter that controls distribution shift -- in your case this is b, in BEAR it is epsilon, in SPIBB it is N_\lambda. So, what is the precise argument about not having concentrability in your bound vs having another factor of horizon? Perhaps a head to head comparison of bounds in the same notation, with the same assumptions, side-by-side would be much better and that needs to be added as well. Also a somewhat minor point: but the paper mentions a couple of times that their bound is better than BEAR (Kumar et al.) but a clear description of why this is the case is missing. I see the high level argument, there's an f(\eps) in those bounds, and here there is no such term, but against \eps is a hyperparameter similar to b, and based on my comments above, I feel like that these things need to be properly explained in a better way. Right now, Lines 219-227 come across as a bit handwavy to me.


Review 2

Summary and Contributions: This paper presents a class of algorithms for batch/offline RL called marginalized behavior supported (MBS) policy iteration and Q iteration. The algorithms attempt to fix an issue with prior work that is not sufficiently conservative by restricting Bellman backups to state, action pairs with high joint probability instead of to actions with high probability conditioned on the state. The paper proves performance bounds for these algorithms competing against the best policy in the policy class with support under the data generating distribution and not depending on a concentrability coefficient. Finally, some toy experiments show improvement over BCQ and SPIBB.

Strengths: 1. The main strength of the paper is the strong and sound theoretical analysis. The analysis replaces the concentrability coefficient by the assumptions of bounded density, access to a good density estimator, and adapted completeness assumptions. This is a strong contribution which extends more than a decade of results using concentrability to analyze AVI and API. 2. The analysis is novel and requires an algorithmic change. This makes the contribution of potential significance to the community by showing that making backups conservative with respect to joint distributions of state, action pairs is more principled than just using the conditional distributions. 3. The toy experiments on the MDP with rare transition and combination lock demonstrate the potential of the proposed algorithm without function approximation.

Weaknesses: 1. The main weakness of this paper is the experimental section. The theoretical contribution may be able to stand on it's own, but if a "practical" version of the algorithm is to be included, it needs to be more rigorous. 2. First, the baselines seem to be undertuned throughout. In the cartpole experiment, the MBS algorithms use a larger set of threshold hyperparameters and this set itself seems to have been chosen with additional experimentation (as indicated in appendix E). Since all the algorithms seem somewhat sensitive to this threshold hyperparameter, this inconsistency potentially invalidates the conclusions. Similarly, in the hopper experiment, the values for BCQ in this paper of around 1000 seem to be substantially lower than the comparable hopper with imperfect demonstrations scores of around 2500 in the original BCQ paper. This is suspicious and at the very least an explanation for this inconsistency is required. 3. Second, unfortunately experimental evaluation of RL algorithms is notoriously noisy. As a result, the conclusions need to be replicated across well more than two of the smallest mujoco environments to be considered substantiated. 4. I am in general somewhat skeptical that fitting a generative model of the stationary distribution over state action pairs with a VAE is doable in large environments with reasonably sized datasets. This may be why the experiments are limited to smaller settings. It would be good to include experimental evidence of the efficacy of the density estimation and how that impacts the algorithm. 5. The paper also overstates some claims which should be removed. For example on line 108 the paper says that "these algorithms often diverge, likely due to the failure of this assumption". Divergence in fitted Q-iteration could also be due to (for example) compounding errors of poor optimization of neural networks and their uncontrolled extrapolations.

Correctness: The theoretical claims seem to be correct, but the empirical method has some issues as explained above.

Clarity: The paper is generally clearly written. A few things that could be improved: 1. A clear and intuitive description of each term in the bound in the theorem could help the reader to interpret the result 2. The algorithm box appears before any of the terms in the algorithm are defined which makes it difficult to understand

Relation to Prior Work: Connections to prior theoretical work are good. Some connections to recent empirical work are missing. For example, a whole class of recent methods like [1] which use a filtered behavior cloning objective are not addressed or compared to. The experiments could benefit from a more rigorous evaluation, for example using the datasets and baselines presented in [2]. [1] Siegel, Noah Y., et al. "Keep doing what worked: Behavioral modelling priors for offline reinforcement learning." *arXiv preprint arXiv:2002.08396* (2020). [2] Fu, Justin, et al. "D4rl: Datasets for deep data-driven reinforcement learning." *arXiv preprint arXiv:2004.07219* (2020).

Reproducibility: Yes

Additional Feedback: As explained above I think that the theoretical contribution is good, but the experimental section is not. I would encourage the authors to either remove the experimental section (and just focus on examples like the combination lock as empirical validation), or to substantially improve the rigor of the experiments. Minor nitpicks: 1. Why does figure 2(a) only compare to SPIBB while 2(b) only compares to BCQ? It seems that both experiments should consider both baselines. 2. The theoretical analysis relies on the samples in the dataset being independent draws from mu. I understand that this is fairly standard, but in practice the trajectories in the dataset will give correlated samples, how would this effect the analysis? ------------------------------------------------------------------------- ------------------------------------------------------------------------- Post-rebuttal update: The authors do an adequate job responding to my central concerns. In particular, they state that they will increase the emphasis on the theoretical contributions and view the experimental section as a proof-of-concept, not an outright claim that this algorithm, as implemented, is the best batch deep RL algorithm out there. As I said in my review, I think that the theoretical contribution of the paper is sound, interesting, and important. I commend the authors for wanting to demonstrate the practical relevance of their primarily theoretical contribution, and I don't want to hold experimental details against them for a primarily theoretical paper. The authors do a good job explaining how they tune the hyper-parameters. I also think that the toy experiments in Figure 2 already provide a decent proof of concept of where we might expect MBS to help. However, I am still a bit confused by the dramatic difference between the results reported in the BCQ paper with those reported in this paper and D4RL. The explanation of performance dropping after 300k is not sufficient. Looking at Figure 2 of the BCQ paper the lower bound of the error bar is at 2000, but the upper bound of the results for BCQ reported in this paper barely go above 1500. This may well be a problem with the reported results in the original BCQ paper, and I guess the state of experimentation in RL is such that this sort of variability is to be expected. Still, this is not very satisfying. Also, the lack of larger scale experiments is a problem if the authors want to make any sort of stronger empirical claims. I will increase my score to 7 to reflect the reframing of the paper and shoring up of some of the issues with the experiments.


Review 3

Summary and Contributions: The paper propose a new conservative update rule that provides improvements and stability in the batch RL setting. Specifically, it removes the common assumption of concentrability at the core of previous algorithms (FQI/FPI) which is unsuitable in most practical settings. Instead, the analysis of the conservative update relies on the support of the batch data and provides bounds with respect to the optimal policy within the supported class.

Strengths: The authors propose a solution that is both simple and well-justified. Its simplicity leads to an easily implementable idea that works well even on high dimensional tasks. Although the idea of a conservative update rule has been explored previously, it hasn’t been analyzed as thoroughly as in this work. Moreover, the method provides better results more consistently across different settings compared to a range of baselines. From a theoretical point fo view, relaxing the assumption of concentrability is innovative and interesting. The authors show that they can still recover a near-optimal policy that lies within a restricted class of policies (where the restriction depends on the batch size and the cut-off coefficient). If the batch of data provides enough support to include the MDP’s optimal policy within the restricted class, it will recover a policy that is near optimal for the MDP.

Weaknesses: Given the fact that the derivation leads an easily implementable solution, it would make a stronger case if the authors could provide experiments for a few more environments. Notably, BCQ evaluates on HalfCheetah as well as Walker2d, which both require relatively little compute to complete. Moreover, the experiment on Hopper is conducted with only 5 seeds. Given the high variance across seeds in deep RL it would seem more appropriate to have somewhere around 10 seeds. In the experiment on CartPole, it would be interesting to see how the different algorithms react to the size of the dataset. Although the analysis provides bounds that are similar to previous algorithms, removing concentrability assumption brings a different cost: a restricted policy set. The authors show that a near-optimal policy within this class is recoverable, but the class itself is highly dependent on the batch of data. What if the policy class only somehow includes very suboptimal policies? Corollary 1 addresses this issue, but relies on the condition that P( \mu(s,a) < 2b | \pi^{\star}) \eq \epsilon, but it is not clear when such a condition would occur for a reasonable \epsilon. In general, a bit more detailed discussion on the bounds would make it more clear, as more terms than usual seem to appear and all of it involves a restricted policy class.

Correctness: It is said that the proposed algorithm is always better than behaviour cloning, but that not seem true. For certain lower values of \epsilon BC seems to perform better

Clarity: The paper is generally well-written. The section about the concentrability (beginning of section 3) would be a stronger argument if more details were given and a better explanation provided on the problems involved.

Relation to Prior Work: The relation to prior work is well documented.

Reproducibility: Yes

Additional Feedback: Questions: -The authors suggest a hard cutoff, which gives the most pessimistic estimate of 0. What if the cutoff was “softer” by defining \zeta(Sara) to be proportional to \hat{\mu)(s,a)? Could bounds be derived for this case? -The derivation assumes that rewards are positive, but in practice rewards could be negative. Would this be a limitation? -When having access to the true behaviour policy, would SPIBB would outperform the proposed approach? -Is there a reason for not including the error bars in the CartPole experiment? -SPIBB is not included in Figure 2a Typos: L86 “given an initial state-action pair s” L176 “from more of the batch dataset” L30 “max in the Bellman operator may pick (s, a) pairs” (wording is a bit strange as the max does not pick states per se) ========================================== After reading the rebuttal, I feel like the authors have done a good effort in answering the reviewers' concerns and have added experimental results. Some of the discussion in the rebuttal should definitely be included in the paper, notably on the restricted set of policies. I would really appreciate if some additional toy experiments could be done in order to understand how their approach affects the resulting policy and how far this one is compared to the optimal policy. In general, the contribution is novel and solid enough and I keep my score as is.

[Author Response · NeurIPS 2020]



Figure 1: **Left**: Same setting as Figure 4, with 15 seeds. **Mid and Right**: Same setting as Figure 3 (left), with error bars and result in low data regime. (Results with $N = 1000, 5000$ are similar and will be included in revision.) Larger set of $b$ for baselines: BCQ: {0, 0.05, 0.1, 0.2}, SPIBB: {1e-4, 5e-4, 1e-3, 5e-3, 1e-2}, MBS: {5e-4, 1e-3, 5e-3}

We thank the reviewers for the excellent feedback that helped us improve the manuscript. Our revision includes (1)
additional experiments to address reviewer's concern about variability and rigor. (2) study of algorithms' behavior as
the amount of data changes, (3) clearer description of the experiment setup, especially how baselines are tuned, (4)
more emphasis on the theoretical contributions. We acknowledge that experiments with larger domains will be nicer,
but note that (1) many related works have only tabular theory with heuristic extensions to function approximation. We
give sound theory for the significantly harder function approximation setting, carefully accounting for all the error
sources and these results are certainly the primary focus of this work. The experiments are intended to be illustrative
proofs-of-concept. (2) Estimating state action visitation distributions is an active research area (e.g. Batch Stationary
Distribution Estimation by J. Wen et al.) and MBS is composable with better estimators. We aim to study MBS in
higher dimensions in a more empirically focused follow up.

**(R1) Same as truncating conditional state-action probabilities in deterministic dynamics?** No! Censoring based
on conditional probabilities can be viewed as MBS with a very naive assumption that $\mu(s)$ is uniform. Using an
empirical estimate $\hat{\mu}(s)$ through counts in discrete and density estimation in continuous domains is much nicer, and we
capture the effect of modeling error through $\epsilon_\mu$ in our bounds. Even for deterministic domains, many $(s, a)$ pairs can
lead to the same $s'$. Section 3 Figure 1(b) is an example where transitions are deterministic but truncating $\mu(a|s)$ can
fail with any reasonable threshold ($\leq 0.5 = \mu_{max}$ in this case).
**(R1) Comparison with Kumar et al.** We discuss this in lines 222-224 (re the $f(\epsilon)$ term). There is no fundamental
difference between the two notions of concentrability, and our improvement is algorithmic, not just in the analysis.
**(R1) Comparison with residual gradient.** We view residual gradient as a different optimization method of the
AVI/API objectives, but still requires concentrability without additional work. Sure MBS is composable with it.

**(R2) Are baselines in the experiments under-tuned?** No! We use the "imperfect-imitation" experiment setting from
the BCQ paper, their code as well as their hyper-parameter ranges. The only difference is that we run BCQ for more
update steps (our BCQ learning curve shows the performance drops a lot after 300k steps). For comparison, the D4RL
paper reports BCQ performance on their "medium" dataset close to our result ($1000 \sim 1500$). This observation shows
that, without reliable stopping criteria for batch RL some non-conservative algorithms can eventually deteriorate. SPIBB
and MBS use the same hyper-parameter ranges. Appendix E.2 may have caused confusion – that separate experiment
for $b$-ablation searches over a larger set of $b$. We re-ran SPIBB for the larger $b$ set, and BCQ with more $b$ values as well
and observe similar curves. We also replicated all experiments across more random seeds and different data regimes
(see the figures above). Trends are the same as those reported in the paper.

**(R3) What policies lie in the restricted policy set?** Yes it's non-trivial to estimate it. However, rather than assuming
that all available policies satisfy concentrability as in prior works, we provide an *adaptive* scheme which competes with
all policies satisfying this assumption. Since the set is always non-trivial for small enough $b$, the algorithm can always
output something sensible (In contrast, we don't know any guarantee of FQI when concentrability coefficient is infinite).
Also our algorithm doesn't need to know $\Pi_C^{all}$. Corollary 1 covers the special case (learning $\pi^\star$) that shows the major
theoretical advance in our result: only requires coverage for $\pi^\star$ instead of coverage for all $\pi \in \Pi$.

**(R1) Can we compare with BEAR?** Yes, for the final version. We saw persistent gradient explosion when running the
authors' code in this BCQ experiment setting and have extensively debugged the issue with the authors. **(R2) D4RL?** If
the included experiments are not adequate proofs-of-concept, we can include D4RL results.

**(R2, R3) Missing baselines (SPIBB/BCQ) in Figure 2?** Each example in Figure 2 is intended to demonstrate the
weakness of one type of algorithm (BCQ/BEAR and SPIBB). In the other figure they will perform as well as MBS, we
include both methods in both figures as per R2 in the revision.

**(R2, R3) I.i.d. and positive rewards assumption?** We (and prior theoretical works) use the i.i.d. assumption for
cleaner analysis. In batch RL, data comes from a fixed Markov chain – so use of standard martingale concentration is
feasible. If the minimum value $V_{min}$ is known, we can set all blocked back-ups to $V_{min}$ for negative reward settings.

We will incorporate other reviewer suggestions in the final version. Please consider raising the score if we addressed
some of the concerns.

[Meta-Review · NeurIPS 2020]

This is a nice paper, with a new idea and strong theoretical backing. The reviews, rebuttal and discussion periods led to a lot of detailed feedback, so I'd encourage the authors to include as much of this as possible in the camera-ready version, and specifically revise for clarity around the points that were unclear to the reviewers in the first submission.